# Fourier Sparse Leverage Scores and Approximate Kernel Learning

**Tamás Erdélyi**
Texas A&M University
terdelyi@math.tamu.edu

**Cameron Musco**
University of Mass. Amherst
cmusco@cs.umass.edu

**Christopher Musco**
New York University
cmusco@nyu.edu

## Abstract

We prove new explicit upper bounds on the leverage scores of Fourier sparse functions under both the Gaussian and Laplace measures. In particular, we study $s$-sparse functions of the form $f(x) = \sum_{j=1}^{s} a_j e^{i\lambda_j x}$ for coefficients $a_j \in \mathbb{C}$ and frequencies $\lambda_j \in \mathbb{R}$. Bounding Fourier sparse leverage scores under various measures is of pure mathematical interest in approximation theory, and our work extends existing results for the uniform measure [Erd17, CP19a]. Practically, our bounds are motivated by two important applications in machine learning:

**1. Kernel Approximation.** They yield a new random Fourier features algorithm for approximating Gaussian and Cauchy (rational quadratic) kernel matrices. For low-dimensional data, our method uses a near optimal number of features, and its runtime is polynomial in the *statistical dimension* of the approximated kernel matrix. It is the first "oblivious sketching method" with this property for any kernel besides the polynomial kernel, resolving an open question of [AKM+17, AKK+20b].

**2. Active Learning.** They can be used as non-uniform sampling distributions for robust active learning when data follows a Gaussian or Laplace distribution. Using the framework of [AKM+19], we provide essentially optimal results for bandlimited and multiband interpolation, and Gaussian process regression. These results generalize existing work that only applies to uniformly distributed data.

## 1 Introduction

Statistical leverage scores have emerged as an important tool in machine learning and algorithms, with applications including randomized numerical linear algebra [DMM06a, Sar06], efficient kernel methods [AM15, MM17, AKM+17, LTOS19, SK19, LHC+20, FSS19, KKP+20], graph algorithms [SS11, KS16], active learning [DWH18, CVSK16, MMY15, AKM+19], and faster constrained and unconstrained optimization [LS15, AKK+20a].

The purpose of these scores is to quantify how large the magnitude of a function in a particular class can be at a *single location*, in comparison to the *average* magnitude of the function. In other words, they measure how "spiky" a function can be. The function class might consist of all vectors $\mathbf{y} \in \mathbb{R}^n$ which can be written as $\mathbf{Ax}$ for a fixed $\mathbf{A} \in \mathbb{R}^{n \times d}$, all degree $q$ polynomials, all functions with bounded norm in some kernel Hilbert space, or (as in this paper) all functions that are $s$-sparse in the Fourier basis. By quantifying *where* and *how much* such functions can spike to large magnitude, leverage scores help us approximate and reconstruct functions via sampling, leading to provably accurate algorithms for a variety of problems.

Formally, for any class $\mathcal{F}$ of functions mapping some domain $\mathcal{S}$ to the complex numbers $\mathbb{C}$, and any probability density $p$ over $\mathcal{S}$, the leverage score $\tau_{\mathcal{F},p}(x)$ for $x \in \mathcal{S}$ is:

$$\tau_{\mathcal{F},p}(x) = \sup_{f \in \mathcal{F} : \|f\|_p^2 \neq 0} \frac{|f(x)|^2 \cdot p(x)}{\|f\|_p^2} \text{ where } \|f\|_p^2 = \int_{y \in \mathcal{S}} |f(y)|^2 \cdot p(y) \, dy. \quad (1)$$

Readers who have seen leverage scores in the context of machine learning and randomized algorithms [SS11, MMY15, DM16] may be most familiar with the setting where $\mathcal{F}$ is the set of all length $n$ vectors (functions from $\{1, \ldots, n\} \to \mathbb{R}$) which can be written as $\mathbf{Ax}$ for a fixed matrix $\mathbf{A} \in \mathbb{R}^{n \times d}$. In this case, $p$ is taken to be a discrete uniform density over indices $1, \ldots, n$, and it is not hard to check that (1) is equivalent to more familiar definitions of "matrix leverage scores".[1]

When $\mathcal{F}$ is the set of all degree $q$ polynomials, the inverse of the leverage scores is known as the *Christoffel function*. In approximation theory, Christoffel functions are widely studied for different densities $p$ (e.g., Gaussian on $\mathbb{R}$ or uniform on $[-1, 1]$) due to their connection to orthogonal polynomials [Nev86]. Recently, they have found applications in active polynomial regression [RW12, HD15, CCM$^+$15, CM17] and more broadly in machine learning [PBV18, LP19].

We study leverage scores for the class of *Fourier sparse functions*. In particular, we define:[2]

$$\mathcal{T}_s = \left\{ f : f(x) = \sum_{j=1}^{s} a_j e^{i\lambda_j x}, a_j \in \mathbb{C}, \lambda_j \in \mathbb{R} \right\}, \tag{2}$$

where each $\lambda_j$ is the frequency of a complex exponential with coefficient $a_j$. For ease of notation we will denote the leverage scores of $\mathcal{T}_s$ for a distribution $p$ as $\tau_{s,p}(x)$ instead of the full $\tau_{\mathcal{T}_s,p}(x)$.

In approximation theory, the Fourier sparse leverage scores have been studied extensively, typically when $p$ is the uniform density on a finite interval [Tur84, Naz93, BE96, Kós08, Lub15, Erd17]. Recently, these scores have also become of interest in algorithms research due to their value in designing sparse recovery and sparse FFT algorithms in the "off-grid" regime [CKPS16, CP19b, CP19a]. They have also found applications in active learning for bandlimited interpolation, Gaussian process regression, and covariance estimation [AKM$^+$19, MM20, ELMM20].

## 1.1 Closed form leverage score bounds

When studying the leverage scores of a function class over a domain $\mathcal{S}$, one of the primary objectives is to determine the scores for all $x \in \mathcal{S}$. This can be challenging for two reasons:

- For finite domains (e.g., functions on $\mathcal{S} = \{1, \ldots, n\}$) it may be possible to directly solve the optimization problem in (1), but doing so is often computationally expensive.
- For infinite domains (e.g., functions on $\mathcal{S} = [-1, 1]$), $\tau_{\mathcal{F},p}(x)$ is itself a function over $\mathcal{S}$, and typically does not have a simple closed form that is amenable to applications.

Both of these challenges are addressed by shifting the goal from *exactly determining* $\tau_{\mathcal{F},p}(x)$ to *upper bounding* the leverage score function. In particular, the objective is to find some function $\bar{\tau}_{\mathcal{F},p}$ such that $\bar{\tau}_{\mathcal{F},p}(x) \geq \tau_{\mathcal{F},p}(x)$ for all $x \in \mathcal{S}$ and $\int_{x \in \mathcal{S}} \bar{\tau}_{\mathcal{F},p}(x) dy$ is as small as possible.

For linear functions over finite domains, nearly tight upper bounds on the leverage scores can be computed more quickly than the true scores [MDMW12, CLM$^+$15]. Over infinite domains, it is possible to prove for some function classes that $\bar{\tau}_{\mathcal{F},p}(x)$ is always less than some fixed value $C$, sometimes called a Nikolskii constant or coherence parameter [HD15, Mig15, AC20]. In other cases, simple closed form expressions can be proven too upper bound the leverage scores. For example, when $\mathcal{F}$ is the class of degree $q$ polynomials and $p$ is uniform on $[-1, 1]$, the (scaled) Chebyshev density $\bar{\tau}_{\mathcal{F},p}(x) = \frac{2(q+1)}{\pi\sqrt{1-x^2}}$ upper bounds the leverage scores [Lor83, AKM$^+$19].

## 1.2 Our results

The main mathematical results of this work are new upper bounds on the leverage scores $\tau_{s,p}(\cdot)$ of the class of $s$-sparse Fourier functions $\mathcal{T}_s$, when $p$ is a Gaussian or Laplace distribution. These bounds extend known results for the uniform distribution, and are proven by leveraging several results from

approximation theory on concentration properties of exponential sums [Tur84, BE95, BE06, Erd17]. We highlight the applicability of our bounds by developing two applications in machine learning:

**Kernel Approximation (Section 3).** We show that our leverage score upper bounds can be used as importance sampling probabilities to give a modified random Fourier features algorithm [RR07] with essentially tight spectral approximation bounds for Gaussian and Cauchy (rational quadratic) kernel matrices. In fact, we give a black-box reduction, proving that an upper bound on the Fourier sparse leverage scores for a distribution $p$ immediately yields an algorithm for approximating kernel matrices with kernel function equal to the Fourier transform of $p$. This reduction leverages tools from randomized numerical linear algebra, in particular column subset selection results [DMM06b, GS12]. We use these results to show that Fourier sparse functions can universally well approximate kernel space functions, and in turn that the leverage scores of these kernel functions can be bounded using our Fourier sparse leverage score bounds.

Our results make progress on a central open question on the power of oblivious sketching methods in kernel approximation: in particular, whether oblivious methods like random Fourier features and TensorSketch [PP13, CP17, PT20] can match the performance of non-oblivious methods like Nyström approximation [GM13, AM15, MM17]. This question was essentially closed for the polynomial kernel in [AKK$^+$20b]. We give a positive answer for Gaussian and Cauchy kernels in one dimension.

**Active Learning (Appendix C).** It is well known that leverage scores can be used in *active sampling methods* to reduce the statistical complexity of linear function fitting problems like polynomial regression or Gaussian process (GP) regression [CP19a, CM17]. The scores must be chosen with respect to the underlying data distribution $\mathcal{D}$ to obtain an accurate function fit under that distribution [PBV18]. Theorems 1 and 2 immediately yield new active sampling results for regression problems involving $s$ arbitrary complex exponentials when the data follows a Gaussian or Laplacian distribution.

While this result may sound specialized, it's actually quite powerful due to recent work of [AKM$^+$19], which gives a black-box reduction from active sampling for Fourier-sparse regression to active sampling for a wide variety of problems in signal processing and Bayesian learning, including bandlimited function fitting and GP regression. Plugging our results into this framework gives algorithms with essentially optimal statistical complexity: the number of samples required depends on a natural statistical dimension parameter of the problem that is tight in many cases.

We note that any future Fourier sparse leverage score bounds proven for different distributions (beyond Gaussian, Laplace, and uniform) would generalize our applications to new kernel matrices and data distributions. Finally, while our contributions are primarily theoretical, we present experiments on kernel sketching in Section 4. We study a 2-D Gaussian process regression problem, representative of typical data-intensive function interpolation tasks, showing that our oblivious sketching method substantially improves on the original random Fourier features method on which it is based [RR07].

## 1.3 Notation

Boldface capital letters denote matrices or quasi-matrices (linear maps from finite-dimensional vector spaces to infinite-dimensional function spaces). Script letters denote infinite-dimensional operators. Boldface lowercase letters denote vectors or vector-valued functions. Subscripts identify the entries of these objects. E.g., $\mathbf{M}_{j,k}$ is the $(j,k)$ entry of matrix $\mathbf{M}$ and $\mathbf{z}_j$ is the $j^{\text{th}}$ entry of vector $\mathbf{z}$. $\mathbf{I}$ denotes the identity matrix. $\preceq$ denotes the Loewner ordering on positive semidefinite (PSD) matrices: $N \preceq M$ means that $M - N$ is PSD. $\mathbf{A}^*$ denotes the conjugate transpose of a vector or matrix.

## 2 Fourier Sparse Leverage Score Bounds

We now state our main leverage score bounds for the Gaussian and Laplace distributions. These theorems are of mathematical interest and form the cornerstone of our applications in kernel learning:

**Theorem 1** (Gaussian Density Leverage Score Bound). *Consider the Gaussian density* $g(x) = \frac{1}{\sigma\sqrt{2\pi}}e^{-x^2/(2\sigma^2)}$ *and let:*

$$\bar{\tau}_{s,g}(x) = \begin{cases} \frac{1}{\sqrt{2}\sigma} \cdot e^{-x^2/(4\sigma^2)} & \text{for } |x| \geq 6\sqrt{2}\sigma \cdot \sqrt{s} \\ \frac{1}{\sqrt{2}\sigma} \cdot e \cdot s & \text{for } |x| \leq 6\sqrt{2}\sigma \cdot \sqrt{s}. \end{cases}$$

*We have* $\tau_{s,g}(x) \leq \bar{\tau}_{s,g}(x)$ *for all* $x \in \mathbb{R}$ *and* $\int_{-\infty}^{\infty} \bar{\tau}_{s,g}(x)\, dx = O(s^{3/2})$.

We do not know if the upper bound of Theorem 1 is tight, but we know it is close. In particular, if $\mathcal{T}_s$ is restricted to any fixed set of frequencies $\lambda_1 > \ldots > \lambda_s$ it is easy to show that the leverage scores integrate to exactly $s$, and the leverage scores of $\mathcal{T}_s$ can only be larger. So no upper bound can improve on $\int_{-\infty}^{\infty} \bar{\tau}_{s,g}(x)\, dx = O(s^{3/2})$ by more than a $O(\sqrt{s})$ factor. Closing this $O(\sqrt{s})$ gap, either by strengthening Theorem 1, or proving a better lower bound would be very interesting.

**Theorem 2** (Laplace Density Leverage Score Bound). *Consider the Laplace density* $z(x) = \frac{1}{\sqrt{2}\sigma} e^{-|x|\sqrt{2}/\sigma}$ *and let:*

$$\bar{\tau}_{s,z}(x) = \begin{cases} \frac{\sqrt{2}}{\sigma} \cdot e^{-|x|\sqrt{2}/(6\sigma)} \text{ for } |x| \geq 9\sqrt{2}\sigma \cdot s \\ \frac{\sqrt{2}}{\sigma} \cdot \frac{e^2 \cdot s}{1+|x|\sqrt{2}/\sigma} \text{ for } |x| \leq 9\sqrt{2}\sigma \cdot s. \end{cases}$$

*We have* $\tau_{s,z}(x) \leq \bar{\tau}_{s,z}(x)$ *for all* $x \in \mathbb{R}$ *and* $\int_{-\infty}^{\infty} \bar{\tau}_{s,z}(x)\, dx = O(s \ln s)$.

Again, we do not know if Theorem 2 is tight, but $\int_{-\infty}^{\infty} \bar{\tau}_{s,z}(x)\, dx = O(s \ln s)$ cannot be improved below $s$. The best known upper bound for the uniform density also integrates to $O(s \ln s)$ [Erd17].

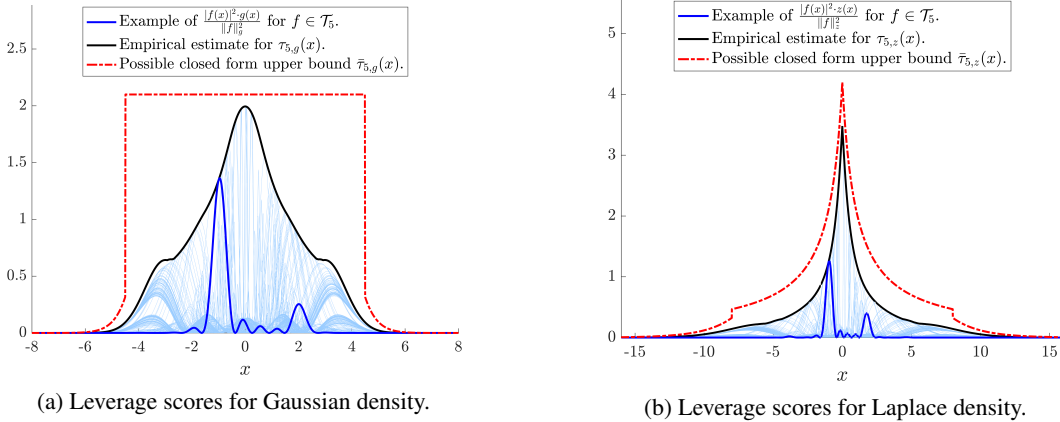

(a) Leverage scores for Gaussian density.

(b) Leverage scores for Laplace density.

Figure 1: Empirically computed (see Appendix D for details) estimates for the Fourier sparse leverage scores, for sparsity $s = 5$. The solid blue lines are normalized magnitudes of 5-sparse Fourier functions that "spike" well above their average. I.e., they plot $|f(x)|^2 \cdot p(x)/\|f\|_p^2$ for various $f \in \mathcal{T}_5$. The leverage score function $\tau_{5,p}(x)$ is the supremum of all such functions. The dashed red lines are closed-form upper bounds for the leverage scores: establishing such bounds is our main research objective. For illustration, the ones plotted here are tighter than what we can currently prove, but they have the same functional form as Theorems 1 and 2 (just with different constants).

Theorems 1 and 2 are proven in Appendix A and the upper bounds visualized in Figure 1. They build on existing results for when $p$ is the uniform distribution over an interval [BE06, Erd17]. This case has been studied since the work of Turán, who proved the first bounds for $\mathcal{T}_s$ and related function classes that are *independent* of the frequencies $\lambda_1, \ldots, \lambda_s$, and only depend on the sparsity $s$ [Tur84, Naz93]. Our bounds take advantage of the exponential form of the Gaussian and Laplace densities $e^{-x^2}$ and $e^{-|x|^2}$. We show how for $f \in \mathcal{T}_s$ to write the weighted function $f(x) \cdot p(x)$ (whose norm under the uniform density equals $f$'s under $p$) in terms of a Fourier sparse function in an extension of $\mathcal{T}_s$ that allows for complex valued frequencies. Combining leverage score type bounds on this extended class [BE06, Erd17] with growth bounds based on Turán's lemma [Tur84, BE95] yields our results.

When the minimum gap between frequencies in $f \in \mathcal{T}_s$ is lower bounded, we also give a tight bound (integrating to $O(s)$) based on Ingham's inequality [Ing36], applicable e.g., in our oblivious embedding results when data points are separated by a minimum distance.

# 3  Kernel Approximation

Given data points[3] $x_1, \ldots, x_n \in \mathbb{R}$ and positive definite kernel function $k : \mathbb{R} \times \mathbb{R} \to \mathbb{R}$, let $\mathbf{K} \in \mathbb{R}^{n \times n}$ be the kernel matrix: $\mathbf{K}_{i,j} = k(x_i, x_j)$ for all $i, j$. $\mathbf{K}$ is the central object in kernel learning methods like kernel regression, PCA, and SVM. Computationally, these methods typically need to invert or find eigenvectors of $\mathbf{K}$, operations that require $O(n^3)$ time. When $n$ is large, this cost is intractable, even for data in low-dimensions. In fact, even the $O(n^2)$ space required to store $\mathbf{K}$ can quickly lead to a computational bottleneck. To address this issue, kernel approximation techniques like random Fourier features methods [RR07], Nyström approximation [WS01, GM13], and TensorSketch [PP13] seek to approximate $\mathbf{K}$ by a low-rank matrix.

These methods compute an explicit embedding $\mathbf{g} : \mathbb{R} \to \mathbb{C}^m$ with $m \ll n$ which can be applied to each data point $x_i$. If $\mathbf{G} \in \mathbb{C}^{m \times n}$ contains $\mathbf{g}(x_i)$ as its $i^{\text{th}}$ column, the goal is for $\tilde{\mathbf{K}} = \mathbf{G}^* \mathbf{G}$, which has rank $m$, to closely approximate $\mathbf{K}$. I.e., for the inner product $\tilde{\mathbf{K}}_{i,j} = \mathbf{g}(x_i)^* \mathbf{g}(x_j)$ to approximate $\mathbf{K}_{i,j}$. If the approximation is good, $\tilde{\mathbf{K}}$ can be used in place of $\mathbf{K}$ in downstream applications. It can be stored in $O(nm)$ space, admits $O(nm)$ time matrix-vector multiplication, and can be inverted exactly in $O(nm^2)$ time, all linear in $n$ when $m$ is small.

**Oblivious Embeddings** Like sketching methods for matrices (see e.g., [Woo14]) kernel approximation algorithms fall into two broad classes.

1. Data *oblivious* methods choose a random embedding $\mathbf{g} : \mathbb{R} \to \mathbb{C}^m$ without looking at the data $x_1, \ldots, x_n$. $\mathbf{g}(x_i)$ can then be applied independently, in parallel, to each data point. Oblivious methods include random Fourier features and TensorSketch methods.

2. Data *adaptive* methods tailor the embedding $\mathbf{g} : \mathbb{R} \to \mathbb{C}^m$ to the data $x_1, \ldots, x_n$. For example, Nyström approximation constructs $\mathbf{g}$ by projecting (in kernel space) each $x_i$ onto $m$ landmark points selected from the data.

Data oblivious methods offer several advantages over adaptive methods: they are easy to parallelize, naturally apply to streaming or dynamic data, and are typically simpler to implement. However, data adaptive methods currently give more accurate kernel approximations than data oblivious methods [MM17]. A major open question in the area [AKM+17, AKK+20b] is if this gap is necessary.

Our main contribution in this section is to establish that it *is not necessary* for the commonly used Gaussian and Cauchy kernels: for low-dimensional data we present a data oblivious method with runtime linear in $n$ that nearly matches the best adaptive methods in speed and approximation quality.

## 3.1  Formal results

Prior work on randomized algorithms for approximating $\mathbf{K}$ considers several metrics of accuracy. We study the following popular approximation guarantee [AM15, MM17, AKK+20b]:

**Definition 1.** *For parameters $\epsilon, \lambda \geq 0$, we say $\tilde{\mathbf{K}}$ is an $(\epsilon, \lambda)$-spectral approximation for $\mathbf{K}$ if:*

$$(1 - \epsilon)(\mathbf{K} + \lambda \mathbf{I}) \preceq \tilde{\mathbf{K}} + \lambda \mathbf{I} \preceq (1 + \epsilon)(\mathbf{K} + \lambda \mathbf{I}). \tag{3}$$

Definition 1 can be used to prove guarantees for downstream applications: e.g., that $\tilde{\mathbf{K}}$ is a good preconditioner for kernel ridge regression with regularization $\lambda$, or that using $\tilde{\mathbf{K}}$ in place of $\mathbf{K}$ leads to statistical risk bounds. See [AKM+17] for details. With (3) as the approximation goal, the data adaptive Nyström method combined with leverage score sampling [AM15] yields the best known kernel approximations among algorithms with runtime linear in $n$. Specifically, for *any* positive semidefinite kernel function the *RLS* algorithm of [MM17] produces an embedding satisfying (3) with $\epsilon = 0$ and with $m = O(s_\lambda \log s_\lambda)$ in $\tilde{O}(n s_\lambda^2)$ time where $s_\lambda$ is the statistical dimension of $\mathbf{K}$:

**Definition 2** ($\lambda$-Statistical Dimension). *The $\lambda$-statistical dimension $s_\lambda$ of a positive semidefinite matrix $\mathbf{K}$ with eigenvalues $\lambda_1 \geq \ldots \geq \lambda_n \geq 0$ is defined as $s_\lambda \stackrel{\text{def}}{=} \sum_{i=1}^{n} \frac{\lambda_i}{\lambda_i + \lambda}$.*

The statistical dimension is a natural complexity measure for approximation $\mathbf{K}$ and the embedding dimension of $O(s_\lambda \log s_\lambda)$ from [MM17] is near optimal.[4] Our main result gives a similar guarantee for two popular kernel functions: the Gaussian kernel $k(x_i, x_j) = e^{-(x_i - x_j)^2/(2\sigma^2)}$ with width $\sigma$ and the Cauchy kernel $k(x_i, x_j) = \frac{1}{1 + (x_i - x_j)^2/\sigma^2}$ with width $\sigma$. The Cauchy kernel is also called the "rational quadratic kernel", e.g., in `sklearn` [PVG+11].

**Theorem 3.** *Consider any set of data points $x_1, \ldots, x_n \in \mathbb{R}$ with associated kernel matrix $\mathbf{K} \in \mathbb{R}^{n \times n}$ which is either Gaussian or Cauchy with arbitrary width parameter $\sigma$. There exists a randomized* oblivious *kernel embedding* $\mathbf{g} : \mathbb{R} \to \mathbb{C}^m$ *such that, if* $\mathbf{G} = [\mathbf{g}(x_1) \ldots, \mathbf{g}(x_n)]$,*with high probability* $\tilde{\mathbf{K}} = \mathbf{G}^* \mathbf{G}$ *satisfies* (3) *with embedding dimension* $m = O(\frac{s_\lambda}{\epsilon^2})$. $\mathbf{G}$ *can be constructed in* $\tilde{O}(n \cdot s_\lambda^{3.5}/\epsilon^4)$ *time for Gaussian kernels and* $\tilde{O}(n \cdot s_\lambda^3/\epsilon^4)$ *time for Cauchy kernels.*

Theorem 3 is a simplified statement of Corollary 26, proven in Appendix B. There we explicitly state the form of $\mathbf{g}$, which as discussed in Section 3.2 below, is composed of a random Fourier features sampling step followed by a standard random projection. For one dimensional data, our method matches the best Nyström method in terms of embedding dimension up to a $1/\epsilon^2$ factor, and in terms of running time up to an $s_\lambda^{1.5}$ factor. It thus provides one of the first nearly optimal oblivious embedding methods for a special class of kernels. The only similar known result applies to polynomial kernels of degree $q$, which can be approximated using the TensorSketch technique [PP13, MSW19, ANW14]. A long line of work on this method culminated in a recent breakthrough achieving embedding dimension $m = O\left(q^4 s_\lambda/\epsilon^2\right)$, with embedding time $O(nm)$ [AKK+20b]. That method can be extended e.g., to the Gaussian kernel, via polynomial approximation of the Gaussian, but one must assume that the data lies within a ball of radius $R$ and the embedding dimension suffers polynomially in $R$.

## 3.2   Our approach

Theorem 3 is based on a modified version of the popular *random Fourier features (RFF)* method from [RR07], and like the original method can be implemented in a few lines of code (see Section 4). As for all RFF methods, it is based on the following standard result for shift-invariant kernel functions:

**Fact 4** (Bochner's Theorem). *For any shift invariant kernel $k(x, y) = k(x - y)$ where $k : \mathbb{R} \to \mathbb{R}$ is a positive definite function with $k(0) = 1$, the inverse Fourier transform given by $p_k(\eta) = \int_{t \in \mathbb{R}} e^{2\pi i \eta t} k(t) dt$ is a probability density function. I.e. $p_k(\eta) \geq 0$ for all $\eta \in \mathbb{R}$ and $\int_{\eta \in \mathbb{R}} p_k(\eta) = 1$.*

As observed by Rahimi and Recht in [RR07], Fact 4 inspires a natural class of linear time randomized algorithms for approximating $\mathbf{K}$. We begin by observing that $\mathbf{K}$ can be written as $\mathbf{K} = \mathbf{\Phi}^* \mathbf{\Phi}$, where $^*$ denotes the Hermitian adjoint and $\mathbf{\Phi} : \mathbb{C}^n \to L_2$ is the linear operator with $[\mathbf{\Phi w}](\eta) = \sqrt{p_k(\eta)} \cdot \sum_{j=1}^n \mathbf{w}_j e^{-2\pi i \eta x_j}$ for $\mathbf{w} \in \mathbb{C}^n, \eta \in \mathbb{R}$.

It is helpful to think of $\mathbf{\Phi}$ as an infinitely tall matrix with $n$ columns and rows indexed by real valued "frequencies" $\eta \in \mathbb{R}$. RFF methods approximate $\mathbf{K}$ by subsampling and reweighting rows (i.e. frequencies) of $\mathbf{\Phi}$ independently at random to form a matrix $\mathbf{G} \in \mathbb{C}^{m \times n}$. $\mathbf{K}$ is approximated by $\tilde{\mathbf{K}} = \mathbf{G}^* \mathbf{G}$. In general, row subsampling is performed using a non-uniform importance sampling distribution. The following general framework for unbiased sampling is described in [AKM+17]:

**Definition 3** (Modified RFF Embedding). *Consider a shift invariant kernel $k : \mathbb{R} \to \mathbb{R}$ with inverse Fourier transform $p_k$. For a chosen PDF $q$ whose support includes that of $p_k$, the Modified RFF embedding $\mathbf{g}(x) : \mathbb{R} \to \mathbb{C}^m$ is obtained by sampling $\eta_1, \ldots, \eta_m$ independently from $q$ and defining:*

$$\mathbf{g}(x) = \frac{1}{\sqrt{m}} \left[ \sqrt{\frac{p_k(\eta_1)}{q(\eta_1)}} e^{-2\pi i \eta_1 x}, \ldots, \sqrt{\frac{p_k(\eta_m)}{q(\eta_m)}} e^{-2\pi i \eta_m x} \right]^*.$$

It is easy to observe that for the modified RFF method $\mathbb{E}[\mathbf{g}(x)^* \mathbf{g}(y)] = k(x, y)$ and thus $\mathbb{E}[\mathbf{G}^* \mathbf{G}] = \mathbf{K}$. So, the feature transformation $\mathbf{g}(\cdot)$ gives an unbiased approximation to $\mathbf{K}$ for any sampling distribution $q$ used to select frequencies. However, a good choice for $q$ is critical in ensuring that

$\mathbf{G}^*\mathbf{G}$ concentrates closely around its expectation with few samples. The original Fourier features method makes the natural choices $q = p_k$, which leads to approximation bounds in terms of $\|\mathbf{K} - \tilde{\mathbf{K}}\|_\infty$ [RR07]. [AKM$^+$17] provides a stronger result by showing that sampling proportional to the so-called *kernel ridge leverage function* is sufficient for an approximation satisfying Definition 1 with $m = O(s_\lambda \log s_\lambda / \epsilon^2)$ samples. That function is defined as follows:

**Definition 4** (Kernel Ridge Leverage Function). *Consider a positive definite, shift invariant kernel $k : \mathbb{R} \to \mathbb{R}$, a set of points $x_1, \ldots, x_n \in \mathbb{R}$ with associated kernel matrix $\mathbf{K} \in \mathbb{R}^{n \times n}$, and a ridge parameter $\lambda \geq 0$. The $\lambda$-ridge leverage score of a frequency $\eta \in \mathbb{R}$ is given by:*

$$\tau_{\lambda,\mathbf{K}}(\eta) = \sup_{\mathbf{w} \in \mathbb{C}^n, \mathbf{w} \neq 0} \frac{|[\mathbf{\Phi}\mathbf{w}](\eta)|^2}{\|\mathbf{\Phi}\mathbf{w}\|_2^2 + \lambda\|\mathbf{w}\|_2^2}.$$

Def. 4 is closely related to the standard leverage score of (1). It measures the worse case concentration of a function $\mathbf{\Phi}\mathbf{w}$ in the span of our kernelized data points at a frequency $\eta$. Since $\|\mathbf{\Phi}\mathbf{w}\|_2^2 = \mathbf{w}^*\mathbf{\Phi}^*\mathbf{\Phi}\mathbf{w} = \mathbf{w}^*\mathbf{K}\mathbf{w}$, leverage score sampling from this class directly aims to preserve $\mathbf{w}^*\mathbf{K}\mathbf{w}$ for worse case $\mathbf{w}$ and thus achieve the spectral guarantee of Def. 1. Due to the additive error $\lambda\mathbf{I}$ in this guarantee, it suffices to bound the concentration with regularization term $\lambda\|\mathbf{w}\|_2^2$ in the denominator.

Of course, the above ridge leverage function is data dependent. To obtain an oblivious sketching method [AKM$^+$17] suggests proving closed form upper bounds on the function, which can be used in its place for sampling. They prove results for the Gaussian kernel, but the bounds require that data lies within a ball of radius $R$, so do not achieve an embedding dimension linear in $s_\lambda$ for any dataset. We improve this result by showing that it is possible to bound the *kernel ridge leverage function* in terms of the *Fourier sparse leverage function* for the density $p_k$ given by the kernel Fourier transform:

**Theorem 5.** *Consider a positive definite, shift invariant kernel $k : \mathbb{R} \to \mathbb{R}$, any points $x_1, \ldots, x_n \in \mathbb{R}$ and the associated kernel matrix $\mathbf{K}$, with statistical dimension $s_\lambda$. Let $s = 6\lceil s_\lambda \rceil + 1$. Then:*

$$\forall \eta \in \mathbb{R}, \quad \tau_{\lambda,\mathbf{K}}(\eta) \leq (2 + 6s_\lambda) \cdot \tau_{s,p_k}(\eta).$$

We prove Theorem 5 in Appendix B. We show that $\mathbf{\Phi}\mathbf{w}$ can be approximated by an $s = 6\lceil s_\lambda \rceil + 1$ Fourier sparse function, so bounding how much it can spike (i.e., which bounds the ridge leverage score of Def. 4) reduces to bounding the Fourier sparse leverage scores. With Theorem 5 in place, we immediately obtain a modified random Fourier features method for any kernel $k$, given an upper bound the Fourier sparse leverage scores of $p_k$. The Fourier transform of the Gaussian kernel is Gaussian, so Theorem 1 provides the required bound. The Fourier transform of the Cauchy kernel is the Laplace distribution, so Theorem 2 provides the required bound.

**Final Embeddings via Random Projection.** In both cases, Theorem 5 combined with our leverage scores bounds does not achieve a tight result alone, yielding embeddings with $m = O(\mathrm{poly}(s_\lambda))$. To achieve the linear dependence on $s_\lambda$ in Theorem 3, we show that it suffices to post-process the modified RFF embedding $\mathbf{g}$ with a standard oblivious random projection method [CNW16]. Proofs are detailed in Appendix B.3, with a complete statement of the random features + random projection embedding algorithm given in Corollary 26.

It is worth noting that, given any approximation $\tilde{\mathbf{K}} = \mathbf{G}^*\mathbf{G}$ satisfying Definition 1, we can always apply oblivious random projection to $\mathbf{G}$ to further reduce the embedding to the target dimension $O\left(\frac{s_\lambda}{\epsilon^2}\right)$, while maintaining the guarantee of Definition 1 up to constants on the error parameters.[5] Thus, the main contribution of Theorem 3 is achieving a lower initial dimension of $\mathbf{G}$ via this sampling step, which directly translates into a faster runtime to produce the final embedding. Our initial embedding dimension, and hence runtime depends polynomially on $s_\lambda$ and $\epsilon$. Existing work [AKM$^+$17, AKK$^+$20b] makes an additional assumption that the data points fall in some radius $R$, and their initial embedding dimension and hence runtime suffers polynomially in this parameter. Related results make no such assumption, but depend linearly on $1/\lambda$ [AKM$^+$17, LTOS19], a quantity which can be much larger than $s_\lambda$ in the typical case when $\mathbf{K}$ has decaying eigenvalues.

# 4 Experimental Results

We now illustrate the potential of Fourier sparse leverage score bounds by empirically evaluating the modified random Fourier features (RFF) method of Section 3. We implement the method without the final JL projection, and use simplifications of the frequency distributions from Theorems 1 and 2, which work well in experiments. For data in $\mathbb{R}^d$ for $d > 1$, we extend these distributions to their natural spherically symmetric versions. See Appendix E for details and Figure 2 for a visualization.

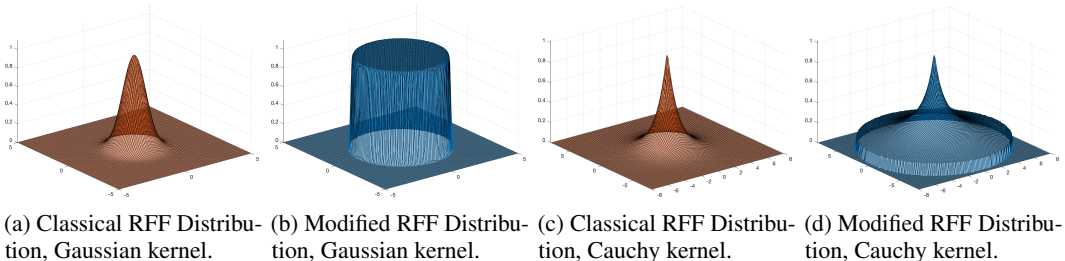

(a) Classical RFF Distribu- (b) Modified RFF Distribu- (c) Classical RFF Distribu- (d) Modified RFF Distribu-
tion, Gaussian kernel.　　　 tion, Gaussian kernel.　　　 tion, Cauchy kernel.　　　　 tion, Cauchy kernel.

Figure 2: Distributions used to sample random Fourier features frequencies $\eta_1, \ldots, \eta_m$. The "Classical RFF" distributions are from the original paper by Rahimi, Recht [RR07]. The "Modified RFF" distributions are simplified versions of the leverage score upper bounds from Thoerems 1 and 2. Notably, our modified distributions sample *high frequencies* (i.e. large $\ell_2$ norm) with higher probability than Classical RFF, leading to theoretical and empirical improvements in kernel approximation.

We compare our method against the classical RFF method on a kernel ridge regression problem involving precipitation data from Slovakia [NM13], a benchmark GIS data set. See Figure 3 for a description. The regression solution requires computing $(\mathbf{K} + \lambda \mathbf{I})^{-1}\mathbf{y}$, where $\mathbf{y}$ is a vector of training data. Doing so with a direct method is slow since $\mathbf{K}$ is large and dense, so an iterative solver is necessary. However, when cross validation is used to choose a kernel width $\sigma$ and regularization parameter $\lambda$, the optimal choices lead to a poorly conditioned system, which leads to slow convergence.

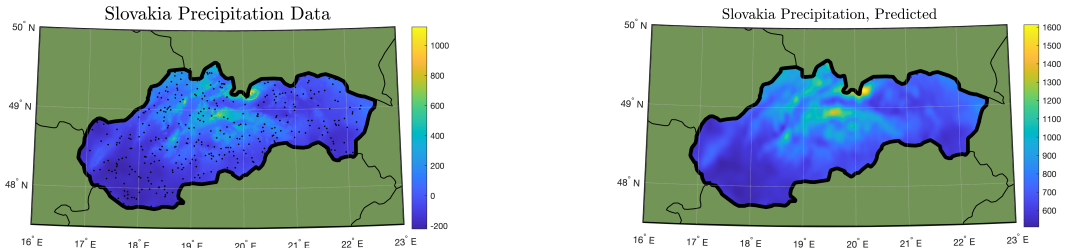

Figure 3: The left image shows precipitation data for Slovakia in mm/year at n = 196k locations on a regular lat/long grid [NM13]. Our goal is to approximate this precipitation function based on 6400 training samples from randomly selected locations (visualized as black dots). The right image shows the prediction given by a kernel regression model with Gaussian kernel, which was computed efficiently using our modified random Fourier method along with a preconditioned CG method.

There are two ways to solve the problem faster using a kernel approximation: either $\tilde{\mathbf{K}}$ can be used in place of $\mathbf{K}$ when solving $(\tilde{\mathbf{K}} + \lambda \mathbf{I})^{-1}\mathbf{y}$, or it can be used as a preconditioner to accelerate the iterative solution of $(\mathbf{K} + \lambda \mathbf{I})^{-1}\mathbf{y}$. We explore the later approach because [AKM$^+$17] already empirically shows the effectiveness of the former. While their modified RFF algorihm is different than ours *in theory*, we both make similar practical simplifications (see Appendix E), which lead our empirically tested methods to be almost identical for the Gaussian kernel. Results on preconditioning are shown in Figure 4. Our modified RFF method leads to substantially faster convergence for a given number of random feature samples, which in turn leads to better downstream prediction error. The superior performance of the modified RFF method can be explained theoretically: our method is designed to target the spectral approximation guarantee of Definition 1, which *is guaranteed to ensure good preconditioning* for $\mathbf{K} + \lambda \mathbf{I}$ [AKM$^+$17]. On the other hand, the classical RFF method actually

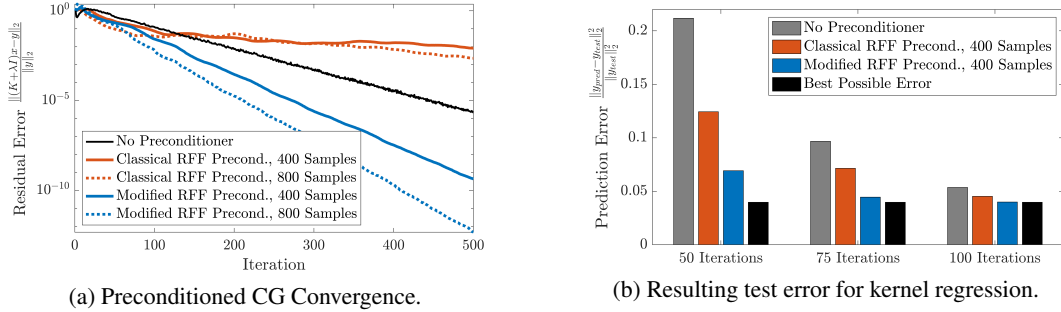

(a) Preconditioned CG Convergence.　　(b) Resulting test error for kernel regression.

Figure 4: The left plot shows residual convergence when solving $\min_{\mathbf{x}} \|(\mathbf{K} + \lambda\mathbf{I})\mathbf{x} - \mathbf{y}\|$ using PCG. Baseline convergence (the black line) is slow, so we preconditioned with both a classical RFF approximation and our modified RFF approximation. Classical RFF accelerates convergence in the high error regime, but slows convergence eventually. Our method significantly accelerates convergence, with better performance as the number of RFF samples increases. On the right, we show that better system solve error leads to better downstream predictions. The black bar represents the relative error of a prediction computed by exactly inverting $\mathbf{K} + \lambda\mathbf{I}$. An approximate solution obtained using our preconditioner approaches this ideal error more rapidly than the other approaches.

achieves better error than our method in other metrics like $\|\mathbf{K} - \tilde{\mathbf{K}}\|_2$, both in theory [Tro15] and empirically (Figure 4). However, for preconditioning, such bounds will not necessarily ensure fast convergence. The key observation is that the spectral guarantee requires better approximation in the *small* eigenspaces of $\mathbf{K}$. By more aggressively sampling higher frequencies that align with these directions (see Figure 2) the modified method obtains a better approximation.

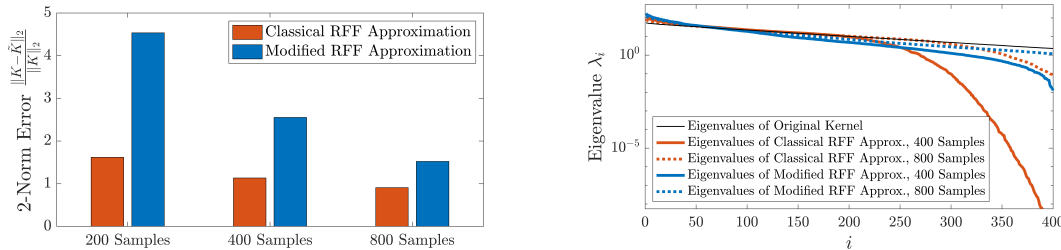

Figure 5: The left plot compares relative spectral norm errors for randomized kernel approximations for a Gaussian kernel matrix $\mathbf{K}$. The classical RFF method actually has *better* error. However, as shown in the right plot, the modified method better approximates the small eigenvalues of $\mathbf{K}$, which is necessary for effective preconditioning as it leads to a better relatively condition number.

## Broader Impacts

Our work contributes to an improved understanding of sampling for kernel approximation and kernel-related function approximation problems. It ties together work in machine learning, signal processing, and approximation theory, which we feel has value in connecting different research communities. Our results in particular focus on low-dimensional interpolation problems, which arise in application areas such as geology, ecology and other scientific fields, medical imaging, and wireless communication. In many of these areas, data driven methods are used to effect positive societal change.

As with all work on efficient learning methods, the algorithms we present, or future variants of them, have the potential to scale inference to even larger data sets than the current state of the art. This can lead to a variety of negative impacts. For example, it may drive the proliferation of massive data collection by corporations and governments for inference tasks, and thus contribute to the associated privacy risks of this data collection. Kernel methods and Gaussian process regression are extremely general tools, used in many applications, including those that may have negative society impacts, such are cell-phone localization, and human and other target tracking. It is possible that our techniques could be employed in these applications.

## Funding Disclosure

We have no external funding to disclose.

## Footnotes

[1]In particular, (1) is equivalent to the definition $\tau_{\mathcal{F},p}(i) = \mathbf{a}_i^T (\mathbf{A}^T \mathbf{A})^{-1} \mathbf{a}_i$ where $\mathbf{a}_i$ is the $i^{\text{th}}$ row of $\mathbf{A}$, and to $\tau_{\mathcal{F},p}(i) = \|u_i\|_2^2$, where $u_i$ is the $i^{\text{th}}$ row of any orthogonal span for $\mathbf{A}$'s columns. See [AKM$^+$17] for details.

[2]It can be observed that any degree $s$ polynomial can be approximated to arbitrarily high accuracy by a function in $\mathcal{T}_s$, by driving the frequencies $\lambda_1, \ldots, \lambda_s$ to zero and taking a Taylor expansion. So the leverage scores of $\mathcal{T}_s$ actually upper bound those of the degree $s$ polynomials [CKPS16].

[3]Results are stated for 1D data, where applications of kernel methods include time series analysis and audio processing. As shown in Section 4, our algorithms easily extend to higher dimensions in practice. In theory, however, extended bounds would likely incur an exponential dependence on dimension, as in [AKM+17].

[4]It can be show that embedding dimension $m = \sum_{i=1}^n \mathbb{1}[\lambda_i \geq \lambda]$ is necessary to achieve (3). Then observe that $s_\lambda \leq \sum_{i=1}^n \mathbb{1}[\lambda_i \geq \lambda] + \frac{1}{\lambda} \sum_{\lambda_i < \lambda} \lambda_i$. For most kernel matrices encountered in practice, the leading term dominates, so $s_\lambda$ is roughly on the order of the optimal $m$.

[5] We also need the slightly stronger condition that $\tilde{\mathbf{K}}$'s statistical dimension is close to that of $\mathbf{K}$. This condition holds for essentially all known sketching methods.

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
