[Supplementary Material]

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

[6]As in [AKM+19], we can generalize the weighted Fourier transform to be weighted by any two measures over $\mathbb{R}$. This allows, for example, the use of discrete measures. We focus on the case when the measures correspond to density functions $p, q$ for simplicity of exposition.

[7]If $\lambda > \|\mathcal{K}_{p,q}\|_{\text{op}}$ then (20) is solved to a constant approximation factor by the trivial solution $\tilde{g} = 0$.

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

# A  Leverage Score Bounds

In this section we give proofs of our Fourier sparse leverage score bounds under the Gaussian and Laplace densities (Theorems 1 and 2). When the minimum gap between frequencies in $f \in \mathcal{T}_s$ is bounded, we also give a tight bound (integrating to $O(s)$) based on Ingham's inequality, applicable e.g., in our oblivious embedding results when data points are separated by a minimum distance.

For notation in this section, we let $\|f\|_2^2 = \int_{x \in \mathbb{R}} |f(x)|^2 dx$ denote the $L_2$ norm of any complex valued function $f : \mathbb{R} \to \mathbb{C}$. We denote the $L_2$ norm over an interval by $\|f\|_{[a,b]}^2 = \int_a^b |f(x)|^2 dx$ and the $L_2$ norm under any density $p$ over $\mathbb{R}$ as $\|f\|_p^2 = \int_{x \in \mathbb{R}} |f(x)|^2 \cdot p(x) \, dx$.

## A.1  Foundational bounds

We build on a number of existing bounds on the uniform density leverage scores and related concentration properties of an extended class of Fourier sparse functions with possibly complex frequencies. This class and its variants have been studied extensively, e.g., in [Tur84, Naz93, BE95, BE96, BE00, Kós08, Lub15, Erd17].

$$\mathcal{E}_s = \left\{ f : f(x) = \sum_{j=1}^s a_j e^{i\lambda_j x}, a_j \in \mathbb{C}, \lambda_j \in \mathbb{C} \right\}. \tag{4}$$

We also consider the subclasses where $\mathcal{E}_s^+$ and $\mathcal{E}_s^-$, which are defined analogously to $\mathcal{E}_s$ but with frequencies $\lambda_j \in \mathbb{C}$ required to have non-negative (respectively, non-positive) real components. Note that our main class of interest $\mathcal{T}_s$ defined in (2) is contained in all three of these extended classes.

We first use a bound on the uniform density leverage score at any point $x$ on an interval, in terms of its distance from the edge of the interval.

**Lemma 6.** *For any $a, b \in \mathbb{R}$ with $a < b$, $x \in (a, b)$, and $f \in \mathcal{E}_s$ with $f \not\equiv 0$:*

$$\frac{|f(x)|^2}{\|f\|_{[a,b]}^2} \leq \frac{s}{\min(x - a, b - x)}.$$

Lemma 6 is stated, up to a constant factor 2 in Theorem 7.1 [Erd17]. We prove it here for completeness and improve this constant.

*Proof.* It is shown in equation (3) of [BE06] that for any $g \in \mathcal{E}_s$ with $g \not\equiv 0$,

$$\frac{|g(0)|^2}{\|g\|_{[-1,1]}^2} \leq s. \tag{5}$$

For $x \in (a, b)$, let $\delta = \min(x - a, b - x)$ and $g(z) = f(x - \delta \cdot z)$. Note that if $f \in \mathcal{E}_s$ and $f \not\equiv 0$, we have $g \in \mathcal{E}_s$ and $g \not\equiv 0$. Additionally, we have $g(0) = f(x)$ and $\|f\|_{[a,b]}^2 \geq \|f\|_{[x-\delta,x+\delta]}^2 = \delta \cdot \|g\|_{[-1,1]}^2$. Applying (5) we then have:

$$\frac{|f(x)|^2}{\|f\|_{[a,b]}^2} \leq \frac{|g(0)|^2}{\delta \cdot \|g\|_{[-1,1]}^2} \leq \frac{s}{\delta},$$

which completes the proof.  $\square$

We note that Lemma 6 can be combined with Lemma 3.2 of [Den16], which tightens bounds proven in [Erd17] and [Kós08] to give the following bound for the uniform density leverage scores:

**Corollary 7** (Uniform Density Leverage Score Bound). *Consider the uniform density $u(x) = \frac{1}{2\sigma}$ for $x \in [-\sigma, \sigma]$, $u(x) = 0$ otherwise, and let*

$$\bar{\tau}_{s,z}(x) = \begin{cases} \frac{s}{\sigma - |x|} & \text{for } |x| \leq \sigma(1 - \frac{4}{\pi s}) \\ \frac{\pi}{4\sigma} s^2 & \text{for } \sigma(1 - \frac{4}{\pi s}) < |x| \leq \sigma \\ 0 & \text{for } |x| > \sigma \end{cases}$$

*We have $\bar{\tau}_{s,u}(x) \geq \tau_{s,u}(x)$ for all $x \in \mathbb{R}$ and $\int_{-\infty}^{\infty} \bar{\tau}_{s,u}(x) \, dx = 2s(1 + \ln(\frac{\pi}{4}s)) = O(s \ln s)$.*

Corollary 7 mirrors our Theorems 1 and 2, and as mentioned in Section 2, no upper bound can improve on the integral of $O(s \ln s)$ by more than a $\ln s$ factor. Understanding if this $\ln s$ can be eliminated or if it is necessary is an interesting open question.

We also employ a bound due to Turán [BE95], which plays a central role in his book [Tur84].

**Lemma 8** (Turán's lemma). *For any $g \in \mathcal{E}_s^+$ and $\alpha, \beta > 0$:*

$$|g(0)| \leq \left( \frac{2e(\alpha + \beta)}{\beta} \right)^s \cdot \|g\|_{[\alpha, \alpha + \beta]}.$$

Turán's lemma can be used to bound the growth of any function in $\mathcal{E}_s^- \supset \mathcal{T}_s$ outside of an interval in terms of its norm on that interval.

**Lemma 9** (Lemma 12.2 [Erd17]). *For any $a \in \mathbb{R}$, $d > 0$, $x \geq a + d$, and $f \in \mathcal{E}_s^-$:*

$$|f(x)| \leq \left( \frac{2e(x - a)}{d} \right)^s \cdot \|f\|_{[a, a+d]}.$$

*Proof.* Let $f \in \mathcal{E}_s^-$. Let $g \in \mathcal{E}_s^+$ be defined by $g(t) := f(x - t)$. We define $\alpha := x - (a + d)$ and $\beta := d$. Applying Lemma 8 with $g \in \mathcal{E}_s^+$ we have

$$|f(x)| = |g(0)| \leq \left( \frac{2e(\alpha + \beta)}{\beta} \right)^s \|g\|_{[\alpha, \alpha + \beta]} = \left( \frac{2e(x - a)}{d} \right)^s \|f\|_{[a, a+d]}.$$

$\square$

Finally, our gap-based result apply to the following restricted class of $\mathcal{T}_s$:

$$\mathcal{T}_{s,\gamma} = \left\{ f : f(x) = \sum_{j=1}^{s} a_j e^{i\lambda_j x}, a_j \in \mathbb{C}, \lambda_j \in \mathbb{R} \text{ with } \min_{j,k} |\lambda_k - \lambda_j| \geq \gamma > 0 \right\}. \quad (6)$$

We denote the leverage score of this class with respect to a density $p$ by $\tau_{s,\gamma,p}(x)$. In bounding these scores we use the following bound due to Ingham [Ing36]:

**Lemma 10** (Ingram's Inequality). *For any $\gamma > 0$, $f \in \mathcal{T}_{s,\gamma}$ with coefficients $a_1, \ldots, a_s$, and $T > \pi/\gamma$,*

$$c_1(T, \gamma) \sum_{j=1}^{s} |a_j|^2 \leq \|f\|_{[-T,T]}^2 \leq c_2(T, \gamma) \sum_{j=1}^{s} |a_j|^2,$$

*where*

$$c_1(T, \gamma) := \frac{4T}{\pi} \left( 1 - \frac{\pi^2}{T^2 \gamma^2} \right) \quad and \quad c_2(T, \gamma) := \frac{16T}{\pi} \left( 1 + \frac{\pi^2}{T^2 \gamma^2} \right).$$

Setting $T = 2\pi/\gamma$ in Ingram's inequality gives:

**Corollary 11.** *For any $\gamma > 0$ and $f \in \mathcal{T}_{s,\gamma}$ with coefficients $a_1, \ldots, a_s$, we have:*

$$\frac{6}{\gamma} \sum_{j=1}^{s} |a_j|^2 \leq \|f\|_{[-2\pi/\gamma, 2\pi/\gamma]}^2 \leq \frac{40}{\gamma} \sum_{j=1}^{s} |a_j|^2.$$

### A.2 Bounds for the Gaussian density

Our leverage score bound for the Gaussian density (Theorem 1) is split into two components – a uniform bound on $\tau_{s,g}(x)$ for all $x \in \mathbb{R}$ (Claim 12) and a bound for $x$ restricted to have sufficiently large magnitude (Claim 13). Combining these two results gives the two part bound of Theorem 1. In this section we focus solely on the unit width Gaussian density: $g(x) = \frac{1}{\sqrt{\pi}} e^{-x^2}$. Bounds under this density can immediately be translated into bounds for any width $\sigma > 0$ via scaling. While they are not applicable to our algorithmic results, we give leverage score lower bounds as well, which help clarify the tightness of the bounds given.

**Claim 12** (Gaussian Leverage Bound – Uniform Bound). *Letting $g(x) = \frac{1}{\sqrt{\pi}}e^{-x^2}$, for all $x \in \mathbb{R}$:*

$$\frac{s}{3\pi} \leq \tau_{\mathcal{E}_s,g} \leq e \cdot s.$$

*As a consequence $\tau_{s,g}(x) \leq e \cdot s$.*

*Proof.* For any $f \in \mathcal{E}_s$ and $a \in \mathbb{R}$, define the shifted and weighted function $w_a(x) = f(x + a) \cdot e^{-(x+a)^2/2}$. We can write:

$$w_a(x) = \sum_{j=1}^{s} a_j e^{i\lambda_j(x+a)} e^{-x^2/2} e^{-a^2/2} e^{-xa}$$

$$= e^{-x^2/2} \cdot \sum_{j=1}^{s} \left( a_j \cdot e^{i\lambda_j a} \cdot e^{-a^2/2} \right) \cdot e^{(i\lambda_j - a)x}.$$

If we let $h_a(x) = \sum_{j=1}^{s} \left( a_j \cdot e^{i\lambda_j a} \cdot e^{-a^2/2} \right) \cdot e^{(i\lambda_j - a)x}$ we thus have $w_a(x) = e^{-x^2/2} \cdot h_a(x)$ and $h_a(x) \in \mathcal{E}_s$. Applying Lemma 6 with $[a, b] = [-1, 1]$ and $x = 0$ gives:

$$\frac{|h_a(0)|^2}{\|h_a\|_{[-1,1]}^2} \leq s.$$

This gives

$$\frac{|w_a(0)|^2}{\|w_a\|_2^2} \leq \frac{|w_a(0)|^2}{\|w_a\|_{[-1,1]}^2} \leq s \cdot \frac{e^0}{e^{-1}} = e \cdot s.$$

Plugging in $a = x$ this gives:

$$\frac{|f(x)|^2 \cdot \frac{1}{\sqrt{\pi}} e^{-x^2}}{\|f\|_g^2} = \frac{|w_x(0)|^2}{\|w_x\|_2^2} \leq e \cdot s.$$

where we use that $\|f\|_g^2 = \frac{1}{\sqrt{\pi}} \|w_a\|_2^2$ for any $a$ due to the weighting $e^{-(x+a)^2/2}$. Thus, we have $\tau_{\mathcal{E}_s,g}(x) \leq e \cdot s$, completing the upper bound.

For the lower bound, let $w_t \in \mathcal{E}_s$ be defined by

$$w_t(x) := f(x - t)e^{tx}, \qquad f(x) := \sum_{j=0}^{s-1} e^{ijx}.$$

We have

$$|w_t(t)|^2 \cdot e^{-t^2} = s^2 e^{2t^2} \cdot e^{-t^2} = s^2 e^{t^2}. \tag{7}$$

Additionally

$$\int_{t\in\mathbb{R}} |w_t(x)|^2 e^{-x^2}\, dx = \int_{x\in\mathbb{R}} |f(x-t)|^2 e^{2tx} e^{-x^2}\, dx$$

$$= e^{t^2} \int_{x\in\mathbb{R}} |f(x-t)|^2 e^{-(x-t)^2}\, dx$$

$$= e^{t^2} \int_{u\in\mathbb{R}} |f(u)|^2 e^{-u^2}\, du.$$

Since $f$ is a sum of complex exponentials with integer frequencies with period $2\pi$, we can bound:

$$\int_{t\in\mathbb{R}} |w_t(x)|^2 e^{-x^2}\, dx \leq e^{t^2} \int_{u\in\mathbb{R}} |f(u)|^2 e^{-u^2}\, du$$

$$\leq e^{t^2} \left( \int_0^\pi |f(u)|^2 e^{-u^2}\, du \right) \cdot \left( 2 + 2\sum_{k=1}^{\infty} e^{-(k\pi)^2} \right)$$

$$\leq e^{t^2} \cdot 3\pi s, \tag{8}$$

where the last bound follows from the fact that $\int_0^\pi |f(x)|^2\, dx = \pi s$. Combining (7) and (8) we obtain the lower bound of the theorem.

$\square$

**Claim 13** (Gaussian Leverage Bound – Large $x$). *Letting $g(x) = \frac{1}{\sqrt{\pi}} e^{-x^2}$, when $|x| \geq 6\sqrt{s}$,*

$$\tau_{s,g}(x) \leq e^{-x^2/2}.$$

*Proof.* Applying Lemma 9 with $a = 0$ and $d = x/2$ gives that for any $f \in \mathcal{T}_s$:

$$\frac{|f(x)|^2}{\|f\|_{[0,x/2]}^2} \leq (4e)^{2s}.$$

This gives in turn that:

$$\tau_{s,g}(x) \leq \frac{e^{-x^2} \cdot (4e)^{2s}}{e^{-(x/2)^2}} \leq e^{-3/4 \cdot x^2 + 6s}. \tag{9}$$

When $|x| \geq 6\sqrt{s}$, $6s \leq \frac{x^2}{6}$ and so (10) gives $\tau_{s,g}(x) \leq e^{(-3/4+1/6) \cdot x^2} \leq e^{-x^2/2}$, completing the claim. $\qquad\square$

We can prove Theorem 1 directly from Claims 12 and 13.

*Proof of Theorem 1.* For the Gaussian density $g(x) = \frac{1}{\sigma\sqrt{2\pi}} \cdot e^{-x^2/(2\sigma^2)}$:

$$\tau_{s,g}(x) = \sup_{f \in \mathcal{T}_s} \frac{|f(x)|^2 \cdot e^{-x^2/(2\sigma^2)}}{\int_{-\infty}^{\infty} |f(y)|^2 e^{-y^2/(2\sigma^2)} \, dy}$$

Let $\bar{g}(x) = \frac{1}{\sqrt{\pi}} e^{-x^2}$ be the Gaussian density with variance $1/2$. For any $f \in \mathcal{T}_s$, let $f_\sigma = f(\sqrt{2}\sigma \cdot x)$. Note that $f_\sigma \in \mathcal{T}_s$. We have:

$$\frac{|f(x)|^2 \cdot e^{-x^2/(2\sigma^2)}}{\int_{-\infty}^{\infty} |f(y)|^2 e^{-y^2/(2\sigma^2)} \, dy} = \frac{|f_\sigma(x/(\sqrt{2}\sigma))|^2 \cdot e^{-x^2/(2\sigma^2)}}{\int_{-\infty}^{\infty} |f_\sigma(y/(\sqrt{2}\sigma))|^2 e^{-y^2/(2\sigma^2)} \, dy} = \frac{|f_\sigma(x/(\sqrt{2}\sigma)|^2 \cdot \bar{g}(x/(\sqrt{2}\sigma))}{\sqrt{2}\sigma \cdot \int_{-\infty}^{\infty} |f_\sigma(y)|^2 \cdot \bar{g}(y) \, dy}.$$

Thus, $\tau_{s,g}(x) = \frac{1}{\sqrt{2}\sigma} \cdot \tau_{s,\bar{g}}(x/(\sqrt{2}\sigma))$. By Claims 12 and 13, if we define:

$$\bar{\tau}_{s,g}(x) = \begin{cases} \frac{1}{\sqrt{2}\sigma} \cdot e^{-x^2/(4\sigma^2)} \text{ for } |x| \geq 6\sqrt{2}\sigma \cdot \sqrt{s} \\ \frac{1}{\sqrt{2}\sigma} \cdot e \cdot s \text{ for } |x| \leq 6\sqrt{2}\sigma \cdot \sqrt{s} \end{cases}$$

we have

$$\tau_{s,g}(x) = \frac{1}{\sqrt{2}\sigma} \cdot \tau_{s,\bar{g}}(x/(\sqrt{2}\sigma)) \leq \bar{\tau}_{s,g}(x).$$

Further,

$$\int_{\infty}^{\infty} \bar{\tau}_{s,g}(x) \, dx = \int_{-6\sqrt{2}\sigma \cdot \sqrt{s}}^{6\sqrt{2}\sigma \cdot \sqrt{s}} \frac{e \cdot s}{\sqrt{2}\sigma} \, dx + \frac{2}{\sigma} \int_{6\sqrt{2}\sigma \cdot \sqrt{s}}^{\infty} e^{-x^2/(4\sigma^2)} \, dx \leq 12es^{3/2} + 1,$$

which completes the theorem. $\qquad\square$

### A.3 Bounds for the Laplace density

We now give bounds for the Laplace density, again focusing on the unit width case and then proving Theorem 2 via a simple scaling argument. Again, our bound is split into two components: a uniform bound for all $x$ and an improved bound for $x$ with large enough magnitude.

**Claim 14** (Laplace Leverage Bound – Universal). *Letting $z(x) = \frac{1}{2} e^{-|x|}$, for all $x \in \mathbb{R}$*

$$\tau_{\mathcal{E}_s,z}(x) \leq \frac{e^2 \cdot s}{1 + |x|}.$$

*As a consequence, $\tau_{s,g}(x) \leq \frac{e^2 \cdot s}{1+|x|}$.*

*Proof.* Assume that $x$ is nonnegative. The same bound holds for negative $x$, since for any $f \in \mathcal{E}_s$, letting $f'(x) = f(-x)$, $f' \in \mathcal{E}_s$ as well. For any $f \in \mathcal{T}_s$ define the weighted function $w(x) = f(x) \cdot \frac{1}{\sqrt{2}} e^{-x/2}$. We can see that $w(x) \in \mathcal{E}_s$ as defined in (4) by writing:

$$w(x) = \frac{1}{\sqrt{2}} \sum_{j=1}^{s} a_j e^{i\lambda_j x} e^{-x/2} = \frac{1}{\sqrt{2}} \sum_{j=1}^{s} a_j e^{(i\lambda_j - 1/2)x}.$$

We define the 'correctly' weighted function $h(x) = f(x) \cdot \frac{1}{\sqrt{2}} e^{-|x|/2}$. Note that for any $y \in [-1, 0]$, we have $h(y) \geq e^{-1} \cdot w(y)$. Thus, we have:

$$\frac{|f(x)|^2 \cdot \frac{1}{2} e^{-|x|}}{\|f\|_z^2} = \frac{|h(x)|^2}{\|h\|_2^2} \leq \frac{|h(x)|^2}{\|h\|_{[-1,2x+1]}^2} \leq e^2 \cdot \frac{|w(x)|^2}{\|w\|_{[-1,2x+1]}^2}.$$

Applying Lemma 6 with $[a, b] = [1, 2x+1]$ then gives:

$$\frac{|f(x)|^2 \cdot \frac{1}{2} e^{-|x|}}{\|f\|_z^2} \leq e^2 \cdot \frac{|w(x)|^2}{\|w\|_{[-1,2x+1]}^2} \leq \frac{e^2 \cdot s}{1 + x},$$

completing the claim. $\qquad \square$

**Claim 15** (Laplace Leverage Bound – Large $x$). *Letting* $z(x) = \frac{1}{2} e^{-|x|}$, *when* $|x| > 18s$,

$$\tau_{s,z}(x) \leq e^{-|x|/6}.$$

*Proof.* The proof is close to that of Claim 13 for the Gaussian density. As in Claim 14, assume without loss of generality that $x$ is nonnegative, so $x > 12s$. Applying Lemma 9 with $a = 0$ and $d = x/2$ gives that for any $f \in \mathcal{T}_s$:

$$\frac{|f(x)|^2}{\|f\|_{[0,x/2]}^2} \leq (4e)^{2s}.$$

This gives:

$$\tau_{s,z}(x) \leq \frac{e^{-x} \cdot (4e)^{2s}}{e^{-x/2}} \leq e^{-x/2+6s}. \tag{10}$$

When $x \geq 18s$, $6s \leq \frac{x}{3}$ and so (10) gives $\tau_{s,g}(x) \leq e^{(-1/2+1/3) \cdot x^2} \leq e^{-x/6}$, completing the claim. $\qquad \square$

We can prove Theorem 2 directly from Claims 15 and 14.

*Proof of Theorem 2.* As in the proof of Theorem 2, we can observe that for the Laplace density $z(x) = \frac{1}{\sqrt{2}\sigma} \cdot e^{-|x|\sqrt{2}/\sigma}$, if we let $\bar{z}(x) = \frac{1}{2} e^{-|x|}$ be the density with variance 2, we have: $\tau_{s,z}(x) = \frac{\sqrt{2}}{\sigma} \cdot \tau_{s,\bar{z}}(x\sqrt{2}/\sigma)$. By Claims 14 and 15, if we define:

$$\bar{\tau}_{s,z}(x) = \begin{cases} \frac{\sqrt{2}}{\sigma} \cdot e^{-|x|\sqrt{2}/(6\sigma)} & \text{for } |x| \geq 9\sqrt{2}\sigma \cdot s \\ \frac{\sqrt{2}}{\sigma} \cdot \frac{e^2 \cdot s}{1+|x|\sqrt{2}/\sigma} & \text{for } |x| \leq 9\sqrt{2}\sigma \cdot s \end{cases}$$

we have

$$\tau_{s,z}(x) = \frac{1}{\sqrt{2}\sigma} \cdot \tau_{s,\bar{z}}(x\sqrt{2}/\sigma) \leq \bar{\tau}_{s,z}(x).$$

Further,

$$\int_\infty^\infty \bar{\tau}_{s,z}(x)\, dx = \frac{2\sqrt{2}e^2}{\sigma} \cdot \int_0^{9\sqrt{2}\sigma s} \frac{s}{1+|x|\sqrt{2}/\sigma}\, dx + \frac{2\sqrt{2}}{\sigma} \int_{9\sqrt{2}\sigma \cdot s}^\infty e^{-x\sqrt{2}/(6\sigma)}\, dx$$

$$= 2e^2 s \cdot \int_0^{18s} \frac{1}{1+x}\, dx + 2 \int_{18s}^\infty e^{-x/6}\, dx$$

$$\leq 2e^2 s \cdot \ln(18s+1) + 1,$$

which completes the theorem. $\qquad \square$

## A.4 Gap-based bounds

Finally, we show how to obtain tighter bounds for the Gaussian density when considering functions in $\mathcal{T}_{s,\gamma}$, whose frequencies have minimum gap $\gamma > 0$ (see (6)). We show:

**Claim 16.** *Letting $g(x) = \frac{1}{\sqrt{\pi}} e^{-x^2}$, for all $x \in \mathbb{R}$:*

$$\tau_{s,\gamma,g}(x) \leq \left(\frac{\gamma}{6} e^{4\pi^2/\gamma^2}\right) \cdot s e^{-x^2}.$$

The above leverage score upper bound is just a scaling of the data density $e^{-x^2}$. For $\gamma = \Omega(1)$, it integrates to $O(s)$, within a constant factor of the lower bound $\int_{x \in \mathbb{R}} \tau_{s,\gamma,g}(x)\, dx \geq s$ given by restricting $\mathcal{T}_{s,\gamma}$ to just just a single fixed set of frequencies.

Claim 16 can be turned into a leverage score bound for the Gaussian density of any width, using the simple scaling argument of Theorem 1 giving:

**Theorem 17** (Gaussian Leverage Bound – Gap Condition)**.** *Consider the Gaussian density with variance $\sigma^2 > 0$, $g(x) = \frac{1}{\sigma\sqrt{2\pi}} e^{-x^2/(2\sigma^2)}$, and let:*

$$\bar{\tau}_{s,\gamma,g}(x) \leq \left(\frac{\gamma}{6} e^{4\pi^2/\gamma^2} \cdot s\right) \cdot \left(\frac{1}{\sqrt{2}\sigma} \cdot e^{-x^2/(2\sigma^2)}\right).$$

*We have $\tau_{s,\gamma,g}(x) \leq \bar{\tau}_{s,\gamma,g}(x)$ for all $x \in \mathbb{R}$ and $\int_{-\infty}^{\infty} \bar{\tau}_{s,\gamma,g}(x)dx = \frac{\gamma}{6} e^{4\pi^2/\gamma^2} \cdot \sqrt{\pi}s$.*

*Proof of Claim 16.* Consider $f \in \mathcal{T}_{s,\gamma}$ with $f(x) := \sum_{j=1}^{s} a_j e^{i\lambda_j x}$, and $\min_{j,k} |\lambda_k - \lambda_j| \geq \gamma > 0$. Combining the Cauchy-Schwarz inequality with Ingham's inequality (Lemma 10), we obtain

$$|f(x)|^2 = \left|\sum_{j=1}^{s} a_j e^{i\lambda_j x}\right|^2 \leq \left(\sum_{j=1}^{s} |e^{i\lambda_j t}|^2\right)\left(\sum_{j=1}^{s} |a_j|^2\right)$$

$$\leq \frac{\gamma s}{6} \int_{-2\pi/\gamma}^{2\pi/\gamma} \left|\sum_{j=1}^{s} a_j e^{i\lambda_j x}\right|^2 dx$$

$$\leq \left(\frac{\gamma}{6} e^{4\pi^2/\gamma^2}\right) s \int_{\mathbb{R}} |f(x)|^2 e^{-x^2} dx.$$

Hence

$$g(x) \cdot |f(x)|^2 \leq \left(\frac{\gamma}{6} e^{4\pi^2/\gamma^2}\right) s e^{-x^2} \cdot \|f\|_g^2,$$

completing the claim. $\qquad\square$

## B Kernel Approximation

As discussed in Section 3, our result on oblivious kernel embedding (Theorem 3) is based on a result from [AKM+17], which shows that strong kernel approximations can be obtained via random Fourier features methods which sample by the kernel ridge leverage scores of Definition 4:

**Theorem 18** (Kernel Embedding via Leverage Score Sampling, [AKM+17])**.** *Let $s_\lambda$ denote the $\lambda$-statistical dimension of $\mathbf{K}$. Given a function $\bar{\tau}_{\lambda,\mathbf{K}}(\eta)$ with:*

$$\bar{\tau}_{\lambda,\mathbf{K}}(\eta) \geq \tau_{\lambda,\mathbf{K}}(\eta) \text{ for all } \eta \in \mathbb{R} \text{ and } T \overset{\text{def}}{=} \int_{\eta \in \mathbb{R}} \bar{\tau}_{\lambda,\mathbf{K}}(\eta) d\eta,$$

*if we apply modified RFF sampling (Definition 3) with density $q(\eta) = \frac{\bar{\tau}_{\lambda,\mathbf{K}}(\eta)}{T}$ and sample size $m = \frac{3T \ln(16 s_\lambda/\delta)}{\epsilon^2}$, then with probability $\geq 1 - \delta$, $\mathbf{G}^*\mathbf{G}$ is an $(\epsilon, \lambda)$-spectral approximation of $\mathbf{K}$.*

## B.1 Kernel leverage score bounds via Fourier sparse approximation

To make use of Theorem 18, we need access to an upper bound $\bar{\tau}_{\lambda,\mathbf{K}}(\eta)$ on the kernel ridge leverage scores. We remark that $\int_{\eta \in \mathbb{R}} \tau_{\lambda,\mathbf{K}}(\eta) d\eta = \text{tr}(\mathbf{K} + \lambda \mathbf{I})^{-1}\mathbf{K}) = s_\lambda$ [AKM$^+$17]. Thus, if $\bar{\tau}_{\lambda,\mathbf{K}}(\eta)$ is a tight bound, Theorem 18 yields an embedding dimension $m = \tilde{O}(s_\lambda/\epsilon^2)$. Our goal is to obtain a nearly tight bound by reducing the problem of bounding $\tau_{\lambda,\mathbf{K}}$ to that of bounding the Fourier sparse leverage score under the density $p_k$ given by the kernel Fourier transform. We prove:

**Theorem 5.** *Consider a positive definite, shift invariant kernel $k : \mathbb{R} \to \mathbb{R}$, any points $x_1, \ldots, x_n \in \mathbb{R}$ and the associated kernel matrix $\mathbf{K}$, with statistical dimension $s_\lambda$. Let $s = 6\lceil s_\lambda \rceil + 1$. Then:*

$$\forall \eta \in \mathbb{R}, \quad \tau_{\lambda,\mathbf{K}}(\eta) \leq (2 + 6s_\lambda) \cdot \tau_{s,p_k}(\eta).$$

As discussed in Section 3, we prove Theorem 5 by first showing that any function $\mathbf{\Phi w}$ in the span of our kernelized data points is well approximated by via an $O(s_\lambda)$ sparse Fourier function.

This Fourier sparse approximation result is based on the well-known fact that any matrix with bounded statistical dimension can be well approximated via projection onto a small subset of rows or columns [DMM06b, GS12, BDMI14]. In particular, we show via a simple reformulation of known results:

**Theorem 19** (Row Subset Selection). *Consider the setting of Theorem 5. For $t = 6 \cdot \lceil s_\lambda \rceil$, there exists a subset of $t$ indices $i_1, \ldots, i_t \subseteq [n]$ and $\mathbf{Z} \in \mathbb{R}^{t \times n}$ such that, letting $\mathbf{\Phi}_t : \mathbb{C}^t \to L_2$ be the operator with $[\mathbf{\Phi}_t \mathbf{w}](\eta) = \sqrt{p_k(\eta)} \cdot \sum_{j=1}^t \mathbf{w}_j e^{-2\pi i \eta x_{i_j}}$ (i.e., the operator containing the $t$ columns of $\mathbf{\Phi}$ corresponding to the indices $i_1, \ldots, i_t$):*

$$\text{tr}(\mathbf{K} - \mathbf{Z}^T \mathbf{\Phi}_t^* \mathbf{\Phi}_t \mathbf{Z}) \leq 3\lambda s_\lambda \text{ and } \mathbf{Z}^T \mathbf{\Phi}_t^* \mathbf{\Phi}_t \mathbf{Z} \preceq \mathbf{K}.$$

*Proof.* Let $\mathbf{B} \in \mathbb{R}^{n \times n}$ be any matrix squareroot of $\mathbf{K}$ with $\mathbf{B}^T\mathbf{B} = \mathbf{K}$. Since $\mathbf{B}^T\mathbf{B} = \mathbf{\Phi}^*\mathbf{\Phi}$ it suffices to prove the existence of a subset of indices $i_1, \ldots, i_t \subseteq [n]$ and a matrix $\mathbf{Z} \in \mathbb{R}^{t \times n}$ such that, letting $\mathbf{B}_t$ contain the columns of $\mathbf{B}$ corresponding to those indices:

$$\text{tr}(\mathbf{K} - \mathbf{Z}^T \mathbf{B}_t^T \mathbf{B}_t \mathbf{Z}) \leq 3\lambda s_\lambda \text{ and } \mathbf{Z}^T \mathbf{B}_t^T \mathbf{B}_t \mathbf{Z} \preceq \mathbf{K}. \tag{11}$$

Let $\mathbf{Z} = \mathbf{B}_t^+ \mathbf{B}$. Letting $\mathbf{P}_t = \mathbf{B}_t \mathbf{B}_t^+$ be the orthogonal projection matrix onto the columns of $\mathbf{B}_t$, we can see that $\mathbf{Z}^T \mathbf{B}_t^T \mathbf{B}_t \mathbf{Z} = \mathbf{B}^T \mathbf{P}_t^2 \mathbf{B} = \mathbf{B}^T \mathbf{P}_t \mathbf{B}$. We first observe that for any $\mathbf{x} \in \mathbb{R}^n$:

$$\mathbf{x}^T \mathbf{Z}^T \mathbf{B}_t^T \mathbf{B}_t \mathbf{Z} \mathbf{x} = \|\mathbf{P}_t \mathbf{B} \mathbf{x}\|_2^2 \leq \|\mathbf{B}\mathbf{x}\|_2^2 = \mathbf{x}^T \mathbf{K} \mathbf{x},$$

which proves that $\mathbf{Z}^T \mathbf{B}_t^T \mathbf{B}_t \mathbf{Z} \preceq \mathbf{K}$, giving the second part of (11). To prove the first part of (11) we employ an optimal column-based matrix reconstruction result [GS12], Theorem 1.1, which shows that there exists a set of $s = 6 \cdot \lceil s_\lambda \rceil$ indices such that:

$$\|\mathbf{B} - \mathbf{B}_t \mathbf{Z}\|_F^2 \leq 1.5 \|\mathbf{B} - \mathbf{B}_{2\lceil s_\lambda \rceil}\|_F^2, \tag{12}$$

where $\mathbf{B}_{2\lceil s_\lambda \rceil}$ is the best rank-$2\lceil s_\lambda \rceil$ approximation to $\mathbf{B}$ (given by projecting $\mathbf{B}$ onto its top $2\lceil s_\lambda \rceil$ singular vectors). Since $\mathbf{B}_t \mathbf{Z}$ is the projection of $\mathbf{B}$ onto the column space of $\mathbf{B}_t$ we can write via the Pythagorean theorem:

$$\|\mathbf{B} - \mathbf{B}_t \mathbf{Z}\|_F^2 = \|\mathbf{B}\|_F^2 - \|\mathbf{B}_t \mathbf{Z}\|_F^2 = \text{tr}(\mathbf{B}^T\mathbf{B}) - \text{tr}(\mathbf{Z}^T \mathbf{B}_t^T \mathbf{B}_t \mathbf{Z}) = \text{tr}(\mathbf{B}^T\mathbf{B} - \mathbf{Z}^T \mathbf{B}_t^T \mathbf{B}_t \mathbf{Z}).$$

Thus, in combination with (12), if we can show $\|\mathbf{B} - \mathbf{B}_{2\lceil s_\lambda \rceil}\|_F^2 \leq 2\lambda s_\lambda$, we will have

$$\text{tr}(\mathbf{B}^T\mathbf{B} - \mathbf{Z}^T \mathbf{B}_t^T \mathbf{B}_t \mathbf{Z}) \leq 3\lambda s_\lambda,$$

yielding the first part of (11) and the theorem. This bound follows from the fact that $\|\mathbf{B} - \mathbf{B}_{\lceil 2s_\lambda \rceil}\|_F^2 = \sum_{i=2\lceil s_\lambda \rceil + 1}^n \lambda_i(\mathbf{K})$. We can apply the following claim, which quantifies the eigenvalue decay of a matrix in terms of its statistical dimension:

**Claim 20.** *For any positive semidefinite $\mathbf{K} \in \mathbb{R}^{n \times n}$ with statistical dimension $s_\lambda$:*

$$\sum_{i=2\lceil s_\lambda \rceil + 1}^n \lambda_i(\mathbf{K}) \leq 2\lambda s_\lambda.$$

*Proof.* Let $I_\lambda$ be the number of eigenvalues of $\mathbf{K}$ that are $\geq \lambda$. We have:

$$s_\lambda = \sum_{i=1}^n \frac{\lambda_i(\mathbf{K})}{\lambda_i(\mathbf{K}) + \lambda} = \sum_{i=1}^{I_\lambda} \frac{\lambda_i(\mathbf{K})}{\lambda_i(\mathbf{K}) + \lambda} + \sum_{i=I_\lambda+1}^n \frac{\lambda_i(\mathbf{K})}{\lambda_i(\mathbf{K}) + \lambda}$$

$$\geq \frac{1}{2} \cdot I_\lambda + \frac{1}{2\lambda} \sum_{i=I_\lambda+1}^n \lambda_i(\mathbf{K}),$$

where the second line follows from that fact that $\lambda_i(\mathbf{K}) \geq \lambda$ for $i \leq I_\lambda$ and $\lambda_i(\mathbf{K}) < \lambda$ for $i > I_\lambda$ Rearranging we have $2\lceil s_\lambda \rceil \geq 2s_\lambda \geq I_\lambda$ and $2s_\lambda \geq \frac{1}{\lambda} \sum_{i=I_\lambda+1}^n \lambda_i(\mathbf{K})$, and in turn:

$$2s_\lambda \geq \frac{1}{\lambda} \sum_{i=2\lceil s_\lambda \rceil+1}^n \lambda_i(\mathbf{K}) \implies 2\lambda s_\lambda \geq \sum_{i=2\lceil s_\lambda \rceil+1}^n \lambda_i(\mathbf{K}).$$

$\square$

Claim 20 directly gives that $\|\mathbf{B} - \mathbf{B}_{2\lceil s_\lambda \rceil}\|_F^2 = \sum_{i=2\lceil s_\lambda \rceil+1}^n \lambda_i(\mathbf{K}) \leq 2\lambda s_\lambda$, completing the proof of Theorem 19. $\square$

*Proof of Theorem 5.* Applying Theorem 19 we can bound the kernel leverage score by breaking the function $\mathbf{\Phi w}$ into its projection onto $\mathbf{\Phi}_t$, which after a change of density is a $t = \lceil 6s_\lambda \rceil$-sparse Fourier function in $\mathcal{T}_t$, and the residual.

$$\tau_{\lambda,\mathbf{K}}(\eta) = \sup_{\mathbf{w} \in \mathbb{C}^n, \mathbf{w} \neq 0} \frac{|[\mathbf{\Phi w}](\eta)|^2}{\|\mathbf{\Phi w}\|_2^2 + \lambda\|\mathbf{w}\|_2^2} \leq \frac{2|[\mathbf{\Phi}_t \mathbf{Zw}](\eta)|^2}{\|\mathbf{\Phi w}\|_2^2 + \lambda\|\mathbf{w}\|_2^2} + \frac{2|[\mathbf{\Phi w}](\eta) - [\mathbf{\Phi}_t \mathbf{Zw}](\eta)|^2}{\|\mathbf{\Phi w}\|_2^2 + \lambda\|\mathbf{w}\|_2^2}$$

$$\leq \frac{2|[\mathbf{\Phi}_t \mathbf{Zw}](\eta)|^2}{\|\mathbf{\Phi w}\|_2^2} + \frac{2|[\mathbf{\Phi w}](\eta) - [\mathbf{\Phi}_t \mathbf{Zw}](\eta)|^2}{\lambda\|\mathbf{w}\|_2^2}. \quad (13)$$

Since by Theorem 19, $\mathbf{Z}^T \mathbf{\Phi}_t^* \mathbf{\Phi}_t \mathbf{Z} \preceq \mathbf{K}$ we have

$$\|\mathbf{\Phi w}\|_2^2 = \mathbf{w}^T \mathbf{Kw} \geq \mathbf{w}^T \mathbf{Z}^T \mathbf{\Phi}_t^* \mathbf{\Phi}_t \mathbf{Zw} = \|\mathbf{\Phi}_t \mathbf{Zw}\|_2^2,$$

which combined with (13) gives:

$$\tau_{\lambda,\mathbf{K}}(\eta) \leq \frac{2|[\mathbf{\Phi}_t \mathbf{Zw}](\eta)|^2}{\|\mathbf{\Phi}_t \mathbf{Zw}\|_2^2} + \frac{2|[\mathbf{\Phi w}](\eta) - [\mathbf{\Phi}_t \mathbf{Zw}](\eta)|^2}{\lambda\|\mathbf{w}\|_2^2}$$

$$\leq 2\tau_{t,p_k}(\eta) + \frac{2|[\mathbf{\Phi w}](\eta) - [\mathbf{\Phi}_t \mathbf{Zw}](\eta)|^2}{\lambda\|\mathbf{w}\|_2^2}. \quad (14)$$

The second bound follows from the fact that $\frac{[\mathbf{\Phi}_t \mathbf{Zw}](\eta)}{\sqrt{p_k(\eta)}} \in \mathcal{T}_t$. It remains to bound the second term of (14). Let $\mathbf{z}(\eta) \in \mathbb{C}^n$ be the vector with $\mathbf{z}(\eta)_j = \left[ e^{-2\pi i \eta x_j} - \sum_{k=1}^t \mathbf{Z}_{k,j} \cdot e^{-2\pi i \eta x_{i_k}} \right] \cdot \sqrt{p_k(\eta)}$. Then we can bound via Cauchy-Schwarz:

$$\frac{|[\mathbf{\Phi w}](\eta) - [\mathbf{\Phi}_s \mathbf{Zw}](\eta)|^2}{\lambda\|\mathbf{w}\|_2^2} = \frac{|\mathbf{z}(\eta)^* \mathbf{w}|^2}{\lambda\|\mathbf{w}\|_2^2} \leq \frac{\|\mathbf{z}(\eta)\|_2^2}{\lambda}. \quad (15)$$

We bound $\|\mathbf{z}(\eta)\|_2^2$ as:

**Claim 21.** *Let* $\mathbf{z}(\eta) \in \mathbb{C}^n$ *be as defined above.* $\|\mathbf{z}(\eta)\|_2^2 \leq \tau_{t+1,p_k}(\eta) \cdot 3\lambda s_\lambda$.

Combining Claim 21 with (14) and (15) yields:

$$\tau_{\lambda,\mathbf{K}}(\eta) \leq 2\tau_{t,p_k}(\eta) + 6\tau_{t+1,p_k}(\eta) \cdot s_\lambda \leq (2 + 6s_\lambda) \cdot \tau_{6t+1,p_k}(\eta),$$

which completes the theorem after recalling that we set $t = \lceil s_\lambda \rceil$ in Theorem 19. $\square$

*Proof of Claim 21.* Consider the function $g_j(\eta) = \mathbf{z}(\eta)_j$ and $g(\eta) = \sum_{j=1}^n |g_j(\eta)|^2 = \|\mathbf{z}(\eta)\|_2^2$.

$$g_j(\eta) = \left[ e^{-2\pi i \eta x_j} - \sum_{k=1}^s \mathbf{Z}(k,j) \cdot e^{-2\pi i \eta x_{i_k}} \right] \cdot \sqrt{p_k(\eta)}$$

and thus, $h(\eta) \stackrel{\text{def}}{=} \frac{g_j(\eta)}{\sqrt{p_k(\eta)}} \in \mathcal{T}_{t+1}$ and so:

$$\frac{|g_j(\eta)|^2}{\|g_j\|_2^2} = \frac{p_k(\eta) \cdot |h(\eta)|^2}{\|h\|_{p_k}^2} \le \tau_{t+1, p_k}(\eta).$$

This gives:

$$\|\mathbf{z}(\eta)\|_2^2 = \sum_{j=1}^n |g_j(\eta)|^2 \le \tau_{t+1, p_k}(\eta) \cdot \sum_{j=1}^n \|g_j\|_2^2$$
$$= \tau_{t+1, p_k}(\eta) \cdot \operatorname{tr}(\mathbf{K} - \mathbf{Z}^T \mathbf{\Phi}_s^* \mathbf{\Phi}_s \mathbf{Z})$$
$$\le \tau_{t+1, p_k}(\eta) \cdot 3 s_\lambda,$$

where the last bound follows from Theorem 19. $\qquad\square$

## B.2 Oblivious kernel embedding via keverage score-based RFF

We finally combine our Fourier sparse leverage score bounds of Theorems 1 and 2 with the kernel ridge leverage score bound of Theorem 5 and the leverage score sampling result of Theorem 18 to give oblivious kernel embedding results for the kernels corresponding to the Fourier transforms of the Gaussian and Laplace densities – i.e., the Gaussian and Cauchy (rational quadratic) kernel.

**Corollary 22** (Modified RFF Embedding – Gaussian Kernel). *Consider any set of points $x_1, \ldots, x_n \in \mathbb{R}$ and the associated Gaussian kernel matrix $\mathbf{K} \in \mathbb{R}^{n \times n}$ with $\mathbf{K}_{i,j} = e^{-(x_i - x_j)^2/(2\sigma^2)}$. Let $s_\lambda$ be the $\lambda$-statistical dimension of $\mathbf{K}$, $s = 6\lceil s_\lambda \rceil + 1$, and $q(\eta)$ be the density proportional to:*

$$q(\eta) \propto \begin{cases} e^{-\eta^2 \cdot \pi^2 \cdot \sigma^2} \text{ for } |\eta| \ge \frac{3\sqrt{2}}{\sigma\pi} \cdot \sqrt{s} \\ e \cdot s \text{ for } |\eta| \le \frac{3\sqrt{2}}{\sigma\pi} \cdot \sqrt{s}. \end{cases}$$

*The modified RFF embedding (Def. 3) with density $q(\eta)$ and sample size $m = O\left( \frac{s_\lambda^{5/2} \cdot \log(s_\lambda/\delta)}{\epsilon^2} \right)$, satisfies $\mathbf{G}^* \mathbf{G}$ is an $(\epsilon, \lambda)$-spectral approximation of $\mathbf{K}$ with probability $\ge 1 - \delta$. The embedding $\mathbf{g}(x_i) \in \mathbb{C}^m$, can be constructed obliviously in $O(m)$ time.*

*Proof.* For the Gaussian kernel with width $\sigma$, the Fourier transform density is also Gaussian with variance $\frac{1}{4\pi^2\sigma^2}$:

$$p_k(\eta) = \int_{t \in \mathbb{R}} e^{2\pi i \eta t} e^{-\frac{t^2}{2\sigma^2}} dt = \sigma\sqrt{2\pi} \cdot e^{-2\sigma^2 \pi^2 \eta^2}.$$

Applying Theorem 5 we have: $\tau_{\lambda, \mathbf{K}}(\eta) \le (2 + 6 s_\lambda) \cdot \tau_{s, p_k}(\eta)$ for $s = 6\lceil s_\lambda \rceil + 1$. In turn, applying Theorem 1 gives $\tau_{\lambda, \mathbf{K}}(\eta) \le \bar{\tau}_{\lambda, \mathbf{K}}(\eta)$ where:

$$\bar{\tau}_{\lambda, \mathbf{K}}(\eta) = \begin{cases} (2 + 6 s_\lambda) \cdot \pi\sqrt{2} \cdot \sigma \cdot e^{-\eta^2 \cdot \pi^2 \cdot \sigma^2} \text{ for } |\eta| \ge \frac{3\sqrt{2}}{\sigma\pi} \cdot \sqrt{s} \\ (2 + 6 s_\lambda) \cdot \pi\sqrt{2} e \cdot \sigma \cdot s \text{ for } |\eta| \le \frac{3\sqrt{2}}{\sigma\pi} \cdot \sqrt{s}. \end{cases}$$

Thus, by Theorem 18, if we let $q(\eta)$ be the density proportional to $\bar{\tau}_{\lambda, \mathbf{K}}(\eta)$, a random Fourier features approximation satisfies the guarantee of the Theorem with sample size $m$ given by:

$$m = O\left( \frac{\int_{\eta \in \mathbb{R}} \bar{\tau}_{\lambda, \mathbf{K}}(\eta) d\eta \cdot \log(s_\lambda/\delta)}{\epsilon^2} \right) = O\left( \frac{s_\lambda^{5/2} \cdot \log(s_\lambda/\delta)}{\epsilon^2} \right),$$

since by Theorem 1, $\int_{\eta \in \mathbb{R}} \bar{\tau}_{\lambda, \mathbf{K}}(\eta) d\eta = (2 + 6 s_\lambda) \cdot O(s^{3/2}) = O(s_\lambda^{5/2})$.

Finally, we observe that $q(\eta)$ is just a mixture of a Gaussian density with a uniform density, and hence can be sampled from in $O(1)$ time. Thus each embedding $\mathbf{g}(x_i) \in \mathbb{C}^m$ can be constructed obliviously in $O(m)$ time. $\qquad\square$

We give a very similar result for the Cauchy (also known as rational quadratic) kernel using our Laplacian distribution leverage score bound of Theorem 2.

**Corollary 23** (Modified RFF Embedding – Cauchy Kernel). *Consider any set of points $x_1, \ldots, x_n \in \mathbb{R}$ and the associated Cauchy kernel matrix $\mathbf{K} \in \mathbb{R}^{n \times n}$ with $\mathbf{K}_{i,j} = \frac{1}{1+(x_i-x_j)^2/\sigma^2}$. Let $s_\lambda$ be the $\lambda$-statistical dimension of $\mathbf{K}$, $s = 6\lceil s_\lambda \rceil + 1$, and $q(\eta)$ be the density proportional to:*

$$q(\eta) \propto \begin{cases} e^{-|\eta| \cdot \sigma\pi/3} \text{ for } |\eta| \geq \frac{9s}{\sigma\pi} \\ \frac{e^2 s}{1+|\eta| \cdot 2\sigma\pi} \text{ for } |\eta| \leq \frac{9s}{\sigma\pi}. \end{cases}$$

*The modified RFF embedding (Def. 3) with density $q(\eta)$ and sample size $m = O\left(\frac{s_\lambda^2 \log(s_\lambda) \cdot \log(s_\lambda/\delta)}{\epsilon^2}\right)$ satisfies $\mathbf{G}^*\mathbf{G}$ is an $(\epsilon, \lambda)$-spectral approximation of $\mathbf{K}$ with probability $\geq 1 - \delta$. The embedding $\mathbf{g}(x_i) \in \mathbb{C}^m$, can be constructed obliviously in $O(m)$ time.*

*Proof.* For the Cauchy kernel with width $\sigma$, the Fourier transform density is a Laplace density:

$$p_k(\eta) = \int_{t \in \mathbb{R}} e^{2\pi i \eta t} \frac{1}{1+(t/\sigma)^2} dt = \sigma\pi \cdot e^{-|\eta| \cdot 2\sigma\pi}.$$

Applying Theorem 5 we have: $\tau_{\lambda,\mathbf{K}}(\eta) \leq (2+6s_\lambda) \cdot \tau_{s,p_k}(\eta)$ for $s = 6\lceil s_\lambda \rceil + 1$. In turn, applying Theorem 2 gives $\tau_{\lambda,\mathbf{K}}(\eta) \leq \bar{\tau}_{\lambda,\mathbf{K}}(\eta)$ where:

$$\bar{\tau}_{\lambda,\mathbf{K}}(\eta) = \begin{cases} (2+6s_\lambda) \cdot 2\sigma\pi \cdot e^{-|\eta| \cdot \sigma\pi/3} \text{ for } |\eta| \geq \frac{9s}{\sigma\pi} \\ (2+6s_\lambda) \cdot 2\sigma\pi \cdot \frac{e^2 s}{1+|\eta| \cdot 2\sigma\pi} \text{ for } |\eta| \leq \frac{9s}{\sigma\pi}. \end{cases}$$

Thus, by Theorem 18, if we let $q(\eta)$ be the density proportional to $\bar{\tau}_{\lambda,\mathbf{K}}(\eta)$, a random Fourier features approximation satisfies the guarantee of the theorem with sample size $m$ given by:

$$m = O\left(\frac{\int_{\eta \in \mathbb{R}} \bar{\tau}_{\lambda,\mathbf{K}}(\eta) d\eta \cdot \log(s_\lambda/\delta)}{\epsilon^2}\right) = O\left(\frac{s_\lambda^2 \log(s_\lambda) \cdot \log(s_\lambda/\delta)}{\epsilon^2}\right),$$

since by Theorem 2, $\int_{\eta \in \mathbb{R}} \bar{\tau}_{\lambda,\mathbf{K}}(\eta) d\eta = (2+6s_\lambda) \cdot O(s \log s) = O(s_\lambda^2 \log s_\lambda)$.

Finally, observe that $q(\eta)$ is just a mixture of a Laplacian density with a density of the form $\frac{1}{1+|\eta| \cdot 2\sigma\pi}$. Both can be sampled from in $O(1)$ time using, e.g., inverse transform sampling. Thus each embedding $\mathbf{g}(x_i)$ can be constructed obliviously in $O(m)$ time. $\square$

### B.3 Final embedding via random projection

Corollaries 22 and 23 give oblivious embeddings into $\text{poly}(s_\lambda)$ dimensions via leverage score-based RFF sampling. These oblivious embeddings can be further compressed via standard oblivious random projection time to give an oblivious embedding algorithm achieving the target dimension, linear in $s_\lambda$. Specifically we apply a stable rank approximate matrix multiplication result from [CNW16]:

**Theorem 24** (Random Projection Spectral Approximation). *For any $\mathbf{Z} \in \mathbb{R}^{n \times s}$ and $\mathbf{M} = \mathbf{Z}\mathbf{Z}^T$ with $\lambda$-statistical dimension $s_\lambda$, if $\mathbf{\Pi} \in \mathbb{R}^{s \times m}$ has independent sub-Gaussian entries with variance $1/m$ for $m = O\left(\frac{s_\lambda + \log(1/\delta)}{\epsilon^2}\right)$, then with probability $\geq 1 - \delta$, $\mathbf{Z}\mathbf{\Pi}\mathbf{\Pi}^T\mathbf{Z}^T$ is an $(\epsilon, \lambda)$-spectral approximation of $\mathbf{M}$.*

A simple example of $\mathbf{\Pi}$ that satisfies the theorem is one with independent $\pm 1/\sqrt{m}$ entries. See [CNW16] for more details on sketching matrices that may be used, including sparse ones.

*Proof.* Let $\mathbf{B} = (\mathbf{M} + \lambda\mathbf{I})^{-1/2}\mathbf{Z}$. To prove the theorem it suffices to show that with probability $\geq 1 - \delta$, $\|\mathbf{B}\mathbf{\Pi}\mathbf{\Pi}^T\mathbf{B}^T - \mathbf{B}\mathbf{B}^T\|_2 \leq \epsilon$ as this gives:

$$-\epsilon\mathbf{I} \preceq \mathbf{B}\mathbf{\Pi}\mathbf{\Pi}^T\mathbf{B}^T - \mathbf{B}\mathbf{B}^T \preceq \epsilon\mathbf{I}$$

$$-\epsilon(\mathbf{M} + \lambda\mathbf{I}) \preceq \mathbf{Z}\mathbf{\Pi}\mathbf{\Pi}^T\mathbf{Z}^T - \mathbf{Z}\mathbf{Z}^T \preceq \epsilon(\mathbf{M} + \lambda\mathbf{I})$$

$$\mathbf{M} - \epsilon(\mathbf{M} + \lambda\mathbf{I}) \preceq \mathbf{Z}\mathbf{\Pi}\mathbf{\Pi}^T\mathbf{Z}^T \preceq \mathbf{M} + \epsilon(\mathbf{M} + \lambda\mathbf{I})$$

$$(1 - \epsilon)(\mathbf{M} + \lambda\mathbf{I}) \preceq \mathbf{Z}\mathbf{\Pi}\mathbf{\Pi}^T\mathbf{Z}^T + \lambda\mathbf{I} \preceq (1 + \epsilon)(\mathbf{M} + \lambda\mathbf{I}),$$

which gives the theorem.

To prove that $\|\mathbf{B\Pi\Pi}^T\mathbf{B}^T - \mathbf{B}\mathbf{B}^T\|_2 \le \epsilon$ with probability $\ge 1 - \delta$ we invoke Theorem 1 of [CNW16], which gives that for our setting of $m$, with probability $\ge 1 - \delta$:

$$\|\mathbf{B\Pi\Pi}^T\mathbf{B}^T - \mathbf{B}\mathbf{B}^T\|_2 \le \epsilon \cdot (\|\mathbf{B}\|_2^2 + \|\mathbf{B}\|_F^2/s_\lambda). \tag{16}$$

We have $\|\mathbf{B}\|_2^2 = \|(\mathbf{M} + \lambda\mathbf{I})^{-1/2}\mathbf{M}(\mathbf{M} + \lambda\mathbf{I})^{-1/2}\|_2 \le 1$. Additionally,

$$\|\mathbf{B}\|_F^2 = \sum_{i=1}^n \lambda_i \left( (\mathbf{M} + \lambda\mathbf{I})^{-1/2}\mathbf{M}(\mathbf{M} + \lambda\mathbf{I})^{-1/2} \right)$$

$$= \sum_{i=1}^n \frac{\lambda_i(\mathbf{M})}{\lambda_i(\mathbf{M}) + \lambda} = s_\lambda,$$

giving that $\|\mathbf{B}\|_F^2/s_\lambda = 1$. Thus, by (16) we have with probability $\ge 1 - \delta$, $\|\mathbf{B\Pi\Pi}^T\mathbf{B}^T - \mathbf{B}\mathbf{B}^T\|_2 \le 2\epsilon$, which completes the theorem after adjusting constants. $\qquad\square$

To apply Theorem 24 to the modified RFF embeddings produced by Corollaries 22 and 23, we must argue that these embeddings preserve statistical dimension. We do this via an extension of Theorem 18. Variants of this type of bound are known in the finite matrix approximation setting (e.g., Lemma 20 of [CMM17]).

**Theorem 25** (Leverage Score Sampling Preserves Kernel Statistic Dimension). *Consider the setting of Theorem 18. Letting $s_\lambda(\mathbf{G}^*\mathbf{G})$ and $s_\lambda(\mathbf{K})$ be the $\lambda$-statistical dimensions of $\mathbf{G}^*\mathbf{G}$ and $\mathbf{K}$ respectively, with probability $\ge 1 - \delta$ we have: $s_\lambda(\mathbf{G}^*\mathbf{G}) \le 4s_\lambda(\mathbf{K})$.*

*Proof.* Following Definition 3, the $j^{th}$ row of $\mathbf{G}$ is given by $\sqrt{\frac{1}{m \cdot q(\eta_j)}} \cdot \phi_{\eta_j}$ where $\phi_{\eta_j} \in \mathbb{C}^n$ has $[\phi_{\eta_j}]_k = e^{-2\pi i \eta_j x_k} \cdot \sqrt{p_k(\eta_j)}$. We can write:

$$s_\lambda(\mathbf{G}^*\mathbf{G}) = \mathrm{tr}(\mathbf{G}^*\mathbf{G}(\mathbf{G}^*\mathbf{G} + \lambda\mathbf{I})^{-1})$$

$$= \mathrm{tr}(\mathbf{G}(\mathbf{G}^*\mathbf{G} + \lambda\mathbf{I})^{-1}\mathbf{G}^*)$$

$$= \frac{1}{m} \sum_{j=1}^m \frac{1}{q(\eta_j)} \cdot \phi_{\eta_j}^*(\mathbf{G}^*\mathbf{G} + \lambda\mathbf{I})^{-1}\phi_{\eta_j}.$$

Assuming that the spectral approximation guarantee of Theorem 18 holds, we have $(\mathbf{G}^*\mathbf{G} + \lambda\mathbf{I})^{-1} \le \frac{1}{1-\epsilon}\phi_{\eta_j}^*(\mathbf{K} + \lambda\mathbf{I})^{-1}\phi_{\eta_j} \le 2\phi_{\eta_j}^*(\mathbf{K} + \lambda\mathbf{I})^{-1}\phi_{\eta_j}$ if $\epsilon \le 1/2$. This gives:

$$s_\lambda(\mathbf{G}^*\mathbf{G}) \le \frac{2}{m} \sum_{j=1}^m \frac{1}{q(\eta_j)}\phi_{\eta_j}^*(\mathbf{K} + \lambda\mathbf{I})^{-1}\phi_{\eta_j} = \frac{2}{m} \sum_{j=1}^m \frac{\tau_{\lambda,\mathbf{K}}(\eta_j)}{q(\eta_j)},$$

where we use that $\tau_{\lambda,\mathbf{K}}(\eta_j) = \phi_{\eta_j}^*(\mathbf{K} + \lambda\mathbf{I})^{-1}\phi_{\eta_j}$. This is well known in the finite-dimensional setting, and was proven in [AKM$^+$17] in the kernel setting. Let $S = \frac{2}{m} \sum_{j=1}^m \frac{\tau_{\lambda,\mathbf{K}}(\eta_j)}{q(\eta_j)}$. From above with probability $\ge 1 - \delta$, we have $s_\lambda(\mathbf{G}^*\mathbf{G}) \le S$. Further:

$$\mathbb{E}[S] = 2\mathbb{E}\left[ \frac{\tau_{\lambda,\mathbf{K}}(\eta_j)}{q(\eta_j)} \right] = 2 \int_{\eta \in \mathbb{R}} \tau_{\lambda,\mathbf{K}}(\eta)d\eta = 2s_\lambda(\mathbf{K}).$$

Additionally, by design we have chosen $q(\eta) = \frac{\bar{\tau}_{\lambda,\mathbf{K}}(\eta)}{T}$ for $T \overset{\text{def}}{=} \int_{\eta \in \mathbb{R}} \bar{\tau}_{\lambda,\mathbf{K}}(\eta)d\eta$ and $\bar{\tau}_{\lambda,\mathbf{K}}(\eta) \ge \tau_{\lambda,\mathbf{K}}(\eta)$. Thus $\frac{\tau_{\lambda,\mathbf{K}}(\eta_j)}{q(\eta_j)} \le T$. So by a standard Hoeffding bound,

$$\Pr[S > 4s_\lambda(\mathbf{K})] \le e^{-2ms_\lambda(\mathbf{K})^2/T^2} \le e^{-2m},$$

since $T = \int_{\eta \in \mathbb{R}} \bar{\tau}_{\lambda,\mathbf{K}}(\eta)d\eta \ge \int_{\eta \in \mathbb{R}} \tau_{\lambda,\mathbf{K}}(\eta)d\eta = s_\lambda(\mathbf{K})$. Finally, since $m = \Omega(\log(1/\delta))$, the bound holds with probability at least $1 - \delta$. Overall, via a union bound, we have with probability $1 - 2\delta$, $s_\lambda(\mathbf{G}^*\mathbf{G}) \le S \le 4s_\lambda(\mathbf{K})$, completing the proof after adjusting constants on $\delta$.

$\qquad\square$

Combining Theorem 24 and 25 with Corollaries 22 and 23 gives:

**Corollary 26** (Oblivious Embedding Full Result). *Consider any set of points $x_1, \ldots, x_n \in \mathbb{R}$ and an associated Gaussian kernel matrix $\mathbf{K} \in \mathbb{R}^{n \times n}$. Let $s_\lambda$ be the $\lambda$-statistical dimension of $\mathbf{K}$, $\mathbf{G} \in \mathbb{R}^{n \times m'}$ be the modified RFF embedding of Corollary 22, and $\mathbf{\Pi} \in \mathbb{R}^{m' \times m}$ have independent sub-Gaussian entries with variance $1/m$. Then for $m' = O\left( \frac{s_\lambda^{5/2} \cdot \log(s_\lambda/\delta)}{\epsilon^2} \right)$ and $m = O\left( \frac{s_\lambda + \log(1/\delta)}{\epsilon^2} \right)$, letting $\mathbf{Z} = \mathbf{G}^* \mathbf{\Pi}$, with probability $\geq 1 - \delta$, $\mathbf{Z}\mathbf{Z}^*$ is an $(\epsilon, \lambda)$-spectral approximation of $\mathbf{K}$. The embedding $\mathbf{z}(x_i) \in \mathbb{C}^m$ can be computed obliviously in $O(m' \cdot m) = \text{poly}(s_\lambda, \log(1/\delta), 1/\epsilon)$ time.*

*The same bound holds for the Cauchy kernel using the RFF embedding of Corollary 23 with the $m' = O\left( \frac{s_\lambda^2 \log(s_\lambda) \cdot \log(s_\lambda/\delta)}{\epsilon^2} \right)$.*

# C   Active Learning

We next consider a general active learning problem that encompasses classic problems in both signal processing and machine learning, including e.g., bandlimited function approximation and active Gaussian process regression. Informally, given the ability to make noisy measurements of some function $f$, the goal is to fit a function $\tilde{f}$ with small deviation from $f$ under some data density $p$, under the assumption that $f$ has Fourier transform constrained according to some frequency density $q$. For example, when $q$ is uniform on a bounded interval, $f$ is bandlimited. When $q$ Gaussian, $f$ obeys a 'soft bandlimit' tending towards using lower frequencies with higher density under $q$.

Throughout this section we use the following notation: for any density $p$ over $\mathbb{R}$ let $L_2(p)$ denote the space of square integrable functions with respect to $p$, i.e., $f$ with $\|f\|_p^2 = \int_{x \in \mathbb{R}} |f(x)|^2 p(x) dx < \infty$. For $f, g \in L_2(p)$ we denote the inner product $\langle f, g \rangle_p \overset{\text{def}}{=} \int_{x \in \mathbb{R}} f(x)^* g(x) p(x) dx$, where $f(x)^*$ is the conjugate transpose of $f(x)$. For a linear operator $\mathcal{M} : L_2(p) \to L_2(q)$ we define the operator norm as $\|\mathcal{M}\|_{\text{op}} \overset{\text{def}}{=} \sup_{f \in L_2(p) : \|f\|_p = 1} \|\mathcal{M}f\|_q$. We define the weighted Fourier transform with respect to data and frequency densities $p$ and $q$ as:

**Definition 5** (Weighted Fourier Transform). *Let $p, q$ be probability densities on $\mathbb{R}$. Define the weighted Fourier transform $\mathcal{F}_{p,q} : L_2(p) \to L_2(q)$ by:*[6]

$$[\mathcal{F}_{p,q} f](\eta) \overset{\text{def}}{=} \int_{\mathbb{R}} f(x) e^{-2\pi i \eta x} p(x) dx. \tag{17}$$

*The adjoint $\mathcal{F}_{p,q}^*$ such that $\langle g, \mathcal{F}_{p,q} f \rangle_q = \langle \mathcal{F}_{p,q}^* g, f \rangle_p$ is the inverse Fourier transform operator:*

$$[\mathcal{F}_{p,q}^* g](x) \overset{\text{def}}{=} \int_{\mathbb{R}} g(\eta) e^{2\pi i \eta x} q(\eta) d\eta. \tag{18}$$

With Definition 5 in place we can formally define our main active regression problem of interest:

**Problem 6** (Active Function Fitting). *Let $p, q$ be probability densities on $\mathbb{R}$ representing data and frequency densities respectively. Suppose a time domain function $y \in L_2(p)$ can be written as $y = \mathcal{F}_{p,q}^* g$ for some frequency domain function $g \in L_2(q)$ and, for any $x \in \text{supp}(p)$, we can query $y(x) + n(x)$ for some fixed noise function $n \in L_2(p)$. Then, for error parameter $\lambda \geq 0$, our goal is to recover, using as few queries as possible, an approximation $\tilde{y} \in L_2(p)$ satisfying:*

$$\|y - \tilde{y}\|_p^2 \leq C\|n\|_p^2 + \lambda\|g\|_q^2, \tag{19}$$

*where $C \geq 1$ is a fixed positive constant.*

The first error term of (19) depends on $\|n\|_p^2$, which in general is necessary since the noise is adversarial. Information theoretically, we might hope to achieve $C = 1$, but we focus on achieving with a small constant factor of this ideal bound. The second term $\lambda\|g\|_q^2$ is also necessary in general: it is higher when $y$'s Fourier energy under the frequency density $q$ is larger, making $y$ harder to learn. By decreasing $\lambda$ we obtain a better approximation, but at the cost of higher sample complexity.

As discussed, Problem 6 captures a wide range of classical function fitting problems. See [AKM$^+$19] for details and an exposition of prior work.

- When $q$ is uniform on an interval $[-F, F]$, $f = \mathcal{F}^*_{p,q} g$ is bandlimited with bandlimit $F$. Thus Problem 6 corresponds bandlimited approximation, which lies at the core of modern signal processing and Nyquist sampling theory [Whi15, Nyq28, Kot33, Sha49]. Typically, this problem is considered over an infinite time horizon with access to infinite samples at a certain rate. Significant work also studies the problem in the finite sample regime, when $p$ is uniform over an interval [LP61, LP62, SP61, XRY01, OR14].

- When $q$ is uniform on an interval $[-F, F]$, $f = \mathcal{F}^*_{p,q} g$ is bandlimited with bandlimit $F$. Thus Problem 6 corresponds bandlimited approximation, which lies at the core of modern signal processing and Nyquist sampling theory [Whi15, Nyq28, Kot33, Sha49]. Typically, this problem is considered over an infinite time horizon with access to infinite samples at a certain rate. Significant work also studies the problem in the finite sample regime, when $p$ is uniform over an interval [LP61, LP62, SP61, XRY01, OR14].

- When $q$ is a general density, Problem 6 is closely related to Gaussian process regression (also kriging/kernel ridge regression) [HS93, RW06, Ste12] over data distribution $p$ with covariance kernel $k_q$ given by the Fourier transform of $q$. $q$ corresponds to the expected power spectral density of a Gaussian process drawn with this covariance kernel. For example, if $q$ is Gaussian, $k_q$ is the Gaussian kernel. If $q$ is Cauchy, $k_q$ is the exponential kernel. If $q$ is a mixture of Gaussians, so is $k_q$, a so-called spectral mixture kernel [WA13].

Related to the last example above, it is not hard to show that Problem 6 can be solved by an infinite dimensional kernel ridge regression problem, where the kernel space corresponds to the class of functions $\mathcal{F}^*_{p,q} w$ for $w \in L_2(q)$ and the input is the noisy function $y + n$.

**Claim 27** (Claim 4 of [AKM$^+$19])**.** *Consider the setting of Problem 6. Let $\tilde{g} \in L_2(q)$ satisfy:*

$$\|\mathcal{F}^*_{p,q} \tilde{g} - (y+n)\|_p^2 + \lambda \|\tilde{g}\|_q^2 \le C \cdot \min_{w \in L_2(q)} \left[ \|\mathcal{F}^*_{p,q} w - (y+n)\|_p^2 + \lambda \|w\|_q^2 \right] \qquad (20)$$

*for some $C \ge 1$. Then*

$$\|y - \mathcal{F}^*_{p,q} \tilde{g}\|_p^2 \le 2C\lambda \|g\|_q^2 + 2(C+1)\|n\|_p^2.$$

*That is, $\tilde{y} = \mathcal{F}^*_{p,q} \tilde{g}$ solves Problem 6 with error parameters $\lambda' = 2C\lambda$ and $C' = 2(C+1)$.*

We note that Claim 4 of [AKM$^+$19] is stated in the case when $p$ is the uniform density on an interval, however the proof is via a simple application of triangle inequality and holds for any density $p$. Throughout this section, we will employ several results from [AKM$^+$19] that are stated in the case when $p$ is uniform on an interval but generalize to any density $p$.

## C.1 Active function fitting via kernel leverage score sampling

Of course, the optimization problem of Claim 27 cannot be solved exactly, as it requires full access to $y + n$ on $\mathrm{supp}(p)$. The key idea is to solve the problem approximately by sampling $x \in \mathrm{supp}(p)$ according to their ridge leverage scores and querying $y$ at these sampled points.

**Definition 7** (Kernel operator ridge leverage function)**.** *For probability densities $p, q$ on $\mathbb{R}$ and ridge parameter $\lambda \ge 0$, define the $\lambda$-ridge leverage function for $x \in \mathbb{R}$ as:*

$$\tau_{p,q,\lambda}(x) = \sup_{\{w \in L_2(q) \, \|w\|_q > 0\}} \frac{p(x) \cdot \left| [\mathcal{F}^*_{p,q} w](x) \right|^2}{\|\mathcal{F}^*_{p,q} w\|_p^2 + \lambda \|w\|_q^2}. \qquad (21)$$

The above ridge leverage scores are closely related to the standard leverage scores of (1), for the class of functions $f = \mathcal{F}^*_{p,q} w$ for $w \in L_2(q)$, which we fit in Problem 6. Intuitively, we hope to sample our function in locations where this class can place significant mass (weighted by the data density $p$), so that we can accurately solve the regression problem of Claim 27.

Typically however, the standard leverage scores of the class $\mathcal{F}^*_{p,q} w$ will be unbounded. For example, when $q$ is uniform on an interval, this is the space of all bandlimited functions, which may be arbitrarily spiky. The ridge scores account for this by including a regularization term involving $\|w\|_q^2$

which controls the energy of the function and in turn, how spiky it can be. As Problem 6 allows error in terms of $\|w\|_q^2$, sampling by these scores still suffices. We note that if $\|w\|_q^2$ were allowed to be unbounded, i.e., if we set $\lambda = 0$, it would be impossible to Problem 6 for most common frequencies densities $q$ with a finite number of samples.

Definition 7 is closely related to Definition 4 with two key differences: 1) the leverage function is defined are over data points $x \in \mathbb{R}$ rather than frequencies $\eta \in \mathbb{R}$ and 2) both the data and frequency domains are continuous, while in Definition 4 the data domain was a discrete set of $n$ points. Notationally, a minor difference is that in Definition 4 the density $p$ is 'baked into' the Fourier operator $\mathbf{\Phi}$ through a weighting of $\sqrt{p(\eta)}$ on each of its rows.

The ridge leverage function of Definition 7 has received recent attention in the machine learning literature [PBV18, LP19, FSS19]. $\mathcal{F}_{p,q}w$ lies in the kernel Hilbert space corresponding to the kernel $k_q$ whose Fourier transform is $q$. $\|w\|_q^2$ is the norm of the function in the kernel Hilbert space. [PBV18] focuses on bounding the leverage function in the limit as $\lambda \to 0$. In this limiting case, the function can be shown to converge to a simple transformation of the data density $p$. It is due to this kernel interpretation, which we will see more clearly in our following bounds, that we use the term *kernel operator ridge leverage function*.

As in the discrete kernel matrix case, the ridge leverage scores integrate to the statistical dimension of the associated kernel operator, which in this case is infinite dimensional.

**Definition 8** (Kernel operator statistical dimension). *For probability densities $p, q$ define the kernel operator $\mathcal{K}_{p,q} : L_2(p) \to L_2(p)$ as $\mathcal{K}_{p,q} = \mathcal{F}_{p,q}^* \mathcal{F}_{p,q}$. The $\lambda$-statistical dimension of $\mathcal{K}_{p,q}$ is defined:*

$$s_{p,q,\lambda} \overset{\text{def}}{=} \operatorname{tr}(\mathcal{K}_{p,q}(\mathcal{K}_{p,q} + \lambda\mathcal{I})^{-1}) = \sum_{i=1}^{\infty} \frac{\lambda_i(\mathcal{K}_{p,q})}{\lambda_i(\mathcal{K}_{p,q}) + \lambda}, \tag{22}$$

*where $\mathcal{I}$ is the identity operator on $L_2(p)$ and $\lambda_i(\mathcal{K}_{p,q})$ is the $i^{th}$ largest eigenvalue of $\mathcal{K}_{p,q}$. By Theorem 5 of [AKM$^+$19], $\int_{x \in \mathbb{R}} \tau_{p,q,\lambda}(x)dx = s_{p,q,\lambda}$.*

The work of [AKM$^+$19] shows that the kernel operator statistical dimension $s_{p,q,\lambda}$ essentially characterizes the sample complexity of Problem 6. Under very mild assumptions (see Section 6 of [AKM$^+$19] for details), they show that any algorithm solving Problem 6 must use $\Omega(s_{p,q,\lambda})$ samples. Conversely, by sampling data points according to the kernel operator ridge leverage score function (Def. 7), one can achieve a sample complexity nearly matching this lower bound:

**Theorem 28** (Approximate regression via leverage function sampling – Theorem 6 of [AKM$^+$19]). *Assume that $\lambda \leq \|\mathcal{K}_{p,q}\|_{\text{op}}$.[7] Consider a function $\bar{\tau}_{p,q,\lambda}$ with $\bar{\tau}_{p,q,\lambda}(x) \geq \tau_{p,q,\lambda}(x)$ for all $x \in \mathbb{R}$ and let $T = \int_{x \in \mathbb{R}} \bar{\tau}_{p,q,\lambda}(x)dx$. Let $m = c \cdot T \cdot (\log T + 1/\delta)$ for sufficiently large fixed constant $c$ and let $x_1, \ldots, x_m$ be time points sampled independently according to density $h(x) \overset{\text{def}}{=} \frac{\bar{\tau}_{p,q,\lambda}(x)}{T}$. For $j \in 1, \ldots, m$, let $w_j = \sqrt{\frac{p(x_j)}{m \cdot h(x_j)}}$. Let $\mathbf{F} : \mathbb{C}^m \to L_2(q)$ be the operator:*

$$[\mathbf{F}\,\mathbf{g}](\eta) = \sum_{j=1}^{m} w_j \cdot \mathbf{g}_j \cdot e^{-2\pi i \eta x_j}$$

*and $\mathbf{y}, \mathbf{n} \in \mathbb{R}^m$ be the vectors with $\mathbf{y}_j = w_j \cdot y(x_j)$ and $\mathbf{n}_j = w_j \cdot n(x_j)$. Let:*

$$\tilde{g} = \arg\min_{w \in L_2(q)} \left[ \|\mathbf{F}^* w - (\mathbf{y} + \mathbf{n})\|_2^2 + \lambda\|w\|_q^2 \right]. \tag{23}$$

*With probability $\geq 1 - \delta$:*

$$\|\mathcal{F}_{p,q}^* \tilde{g} - (y + n)\|_p^2 + \lambda\|\tilde{g}\|_q^2 \leq 3 \min_{w \in L_2(q)} \left[ \|\mathcal{F}_{p,q}^* w - (y + n)\|_p^2 + \lambda\|w\|_q^2 \right]. \tag{24}$$

Note that via Claim 27, $\mathcal{F}_{p,q}^* \tilde{g}$ of Theorem 28 thus solves Problem 6 with probability $\geq 1 - \delta$ and with error parameters $\lambda' = 6\lambda$ and $C' = 8$. If $\bar{\tau}_{p,q,\lambda}(x)$ is a tight upper bound on the leverage scores, the sample complexity is near linear in $s_{p,q,\lambda} = \int_{x \in \mathbb{R}} \tau_{p,q,\lambda}(x)dx$.

Also note that the subsampled optimization problem of (23) is just a standard kernel ridge regression problem, and thus efficiently solvable. Specifically:

**Claim 29.** *Consider the set up of Theorem 28. Let $k_q : \mathbb{R} \times \mathbb{R} \to \mathbb{R}$ be the shift-invariant kernel with Fourier transform $q$. Let $\mathbf{K} \in \mathbb{R}^{m \times m}$ have $\mathbf{K}_{i,j} = w_i \cdot w_j \cdot k_q(x_i, x_j)$. Then $\tilde{f} = \mathcal{F}_{p,q}^* \tilde{g}$ is given by $\tilde{f}(x) = \mathbf{k}(x)^T \mathbf{z}$ where $\mathbf{z} = (\mathbf{K} + \lambda \mathbf{I})^{-1}(\mathbf{y} + \mathbf{n})$ and $\mathbf{k}(x) = [w_1 \cdot k_q(x_1, x), \ldots, w_m \cdot k_q(x_m, x)]$.*

## C.2   Kernel operator leverage score bound via Fourier sparse approximation

In sum, Theorem 28, combined with Claim 29 and Claim 27 let us solve the active function fitting problem (Problem 6) with near optimal sample complexity, if we can find a sampling distribution $\bar{\tau}_{p,q,\lambda}$ that tightly upper bounds the true kernel operator leverage function $\tau_{p,q,\lambda}$. In this section we show how to do this using the Fourier sparse leverage score bounds of Theorem 1 and 2. We give a bound based on approximating any function $\mathcal{F}_{p,q}^* w$ via a Fourier sparse function, with sparsity linear in the statistical dimension $s_{p,q,\lambda}$. In particular, we prove the following analog to Theorem 5:

**Theorem 30** (Kernel operator leverage function bound). *Let $s = \lceil 36 \cdot s_{p,q,\lambda} \rceil + 1$. For all $x \in \mathbb{R}$:*

$$\tau_{p,q,\lambda}(x) \leq (2 + 8s_{p,q,\lambda}) \cdot \tau_{s,p}(x).$$

In proving Theorem 30, we use the following continuous analog of Theorem 19:

**Theorem 31** (Frequency subset selection – Theorem 9 of [AKM+19] ). *For some $s \leq \lceil 36 \cdot s_{p,q,\lambda} \rceil$ there exists a set of frequencies $\eta_1, \ldots, \eta_s \in \mathbb{C}$ such that, letting $\mathbf{C}_s : \mathbb{C}^s \to L_2(p)$ be the operator $[\mathbf{C}_s \mathbf{w}](x) = \sum_{j=1}^s \mathbf{w}_j e^{-2\pi i \eta_j x}$ and $\mathbf{Z} : L_2(q) \to \mathbb{C}^s$ be the operator $\mathbf{Z} = (\mathbf{C}_s^* \mathbf{C}_s)^{-1} \mathbf{C}_s^* \mathcal{F}_{p,q}$,*

$$\operatorname{tr}(\mathcal{K}_{p,q} - \mathbf{C}_s \mathbf{Z} \mathbf{Z}^* \mathbf{C}_s^*) \leq 4\lambda \cdot s_{p,q,\lambda} \text{ and } \mathbf{Z}^* \mathbf{C}_s^* \mathbf{C}_s \mathbf{Z} \preceq \mathcal{F}_{p,q} \mathcal{F}_{p,q}^*. \tag{25}$$

*Letting $f_x \in L_2(q)$ be given by $f_x(\eta) = e^{2\pi i x \eta}$ and $\mathbf{c}_x \in \mathbb{C}^s$ have $j^{th}$ entry $[\mathbf{c}_x]_j = e^{-2\pi i \eta_j x}$ we can write:* $\operatorname{tr}(\mathcal{K}_{p,q} - \mathbf{C}_s \mathbf{Z} \mathbf{Z}^* \mathbf{C}_s^*) = \int_{x \in \mathbb{R}} \|f_x - \mathbf{Z}^* \mathbf{c}_x\|_q^2 \cdot p(x) dx$.

*Proof of Theorem 30.*   The proof closely follows that of Theorem 5. We can bound the ridge leverage function of Definition 7 by:

$$\tau_{p,q,\lambda}(x) = \sup_{w \in L_2(q), \|w\|_q > 0} \frac{p(x) \cdot |[\mathcal{F}_{p,q}^* w](x)|^2}{\|\mathcal{F}_{p,q}^* w\|_p^2 + \lambda \|w\|_q^2} \tag{26}$$

$$\leq \frac{2p(x) \cdot |[\mathbf{C}_s \mathbf{Z} w](x)|^2}{\|\mathcal{F}_{p,q}^* w\|_p^2} + \frac{2p(x) \cdot |[\mathcal{F}_{p,q}^* w](x) - [\mathbf{C}_s \mathbf{Z} w](x)|^2}{\lambda \|w\|_q^2}. \tag{27}$$

Since by Theorem 30, $\mathbf{Z}^* \mathbf{C}_s^* \mathbf{C}_s \mathbf{Z} \preceq \mathcal{F}_{p,q} \mathcal{F}_{p,q}^*$ we have

$$\|\mathcal{F}_{p,q}^* w\|_p^2 = \langle \mathcal{F}_{p,q}^* w, \mathcal{F}_{p,q}^* w \rangle_p = \langle \mathcal{F}_{p,q} \mathcal{F}_{p,q}^* w, w \rangle_q \geq \langle \mathbf{Z}^* \mathbf{C}_s^* \mathbf{C}_s \mathbf{Z} w, w \rangle_q = \|\mathbf{C}_s \mathbf{Z} w\|_p^2,$$

which combined with (13) gives:

$$\tau_{p,q,\lambda}(x) \leq \frac{2p(x) \cdot |[\mathbf{C}_s \mathbf{Z} w](x)|^2}{\|\mathbf{C}_s \mathbf{Z} w\|_p^2} + \frac{2p(x) \cdot |[\mathcal{F}_{p,q}^* w](x) - [\mathbf{C}_s \mathbf{Z} w](x)|^2}{\lambda \|w\|_q^2}.$$

We can observe that $\mathbf{C}_s \mathbf{Z} w$ is an $\lceil 36 \cdot s_{p,q,\lambda} \rceil = s - 1$ sparse Fourier function in $\mathcal{T}_{s_1}$, giving:

$$\tau_{p,q,\lambda}(x) \leq 2\tau_{s,p}(x) + \frac{2p(x) \cdot |[\mathcal{F}_{p,q}^* w](x) - [\mathbf{C}_s \mathbf{Z} w](x)|^2}{\lambda \|w\|_q^2}. \tag{28}$$

It thus remains to bound the second term. Let $\mathbf{c}_x \in \mathbb{C}^s$ have $j^{th}$ entry $[\mathbf{c}_x]_j = e^{-2\pi i \eta_j x}$. $\mathbf{c}_x$ is the 'row' of the operator $\mathbf{C}$ corresponding to $x$ and we have $[\mathbf{C}_s \mathbf{Z} w](x) = \mathbf{c}_x^T \mathbf{Z} w$. Similarly, let $f_x \in L_2(q)$ be given by $f_x(\eta) = e^{2\pi i \eta x}$. We can write:

$$|[\mathcal{F}_{p,q}^* w](x) - [\mathbf{C}_s \mathbf{Z} w](x)|^2 = |\langle f_x - \mathbf{Z}^* \mathbf{c}_x, w \rangle_q|^2 \leq \|f_x - \mathbf{Z}^* \mathbf{c}_x\|_q^2 \cdot \|w\|_q^2$$

via Cauchy-Schwarz. Plugging back into (28) gives

$$\tau_{p,q,\lambda}(x) \leq 2\tau_{s,p}(x) + \frac{2p(x) \cdot \|f_x - \mathbf{Z}^* \mathbf{c}_x\|_q^2}{\lambda}. \tag{29}$$

The theorem then follows from (29) combined with the following claim:

**Claim 32.** $p(x) \cdot \|f_x - \mathbf{Z}^* \mathbf{c}_x\|_q^2 \leq \tau_{s,p}(x) \cdot 4\lambda s_{p,q,\lambda}$.

*Proof.* Let $f_\eta \in L_2(p)$ be given by $f_\eta(x) = e^{2\pi i \eta x}$ and let $\mathbf{z}_\eta \in \mathbb{C}^s$ be the 'column' of $\mathbf{Z}$ corresponding to $\eta$. Formally, as $\mathbf{Z} = (\mathbf{C}_s^* \mathbf{C}_s)^{-1} \mathbf{C}_s^* \mathcal{F}_{p,q}^*$, $\mathbf{z}_\eta = (\mathbf{C}_s^* \mathbf{C}_s)^{-1} \mathbf{C}_s^* f_\eta$. We have $f_\eta - \mathbf{C}_s \mathbf{z}_\eta \in \mathcal{T}_s$ and thus:

$$\frac{p(x) \cdot |f_\eta(x) - [\mathbf{C}_s \mathbf{z}_\eta](x)|^2}{\|f_\eta - \mathbf{C}_s \mathbf{z}\|_p^2} \leq \tau_{s,p}(x).$$

This gives:

$$p(x) \cdot \int_{\eta \in \mathbb{R}} |f_\eta(x) - [\mathbf{C}_s \mathbf{z}_\eta](x)|^2 q(\eta) d\eta \leq \tau_{s,p}(x) \cdot \int_{\eta \in \mathbb{R}} \|f_\eta - \mathbf{C}_s \mathbf{z}_\eta\|_p^2 q(\eta) d\eta.$$

Note that $f_\eta(x) = f_x(\eta)$ and $\mathbf{C}_s \mathbf{z}_\eta(x) = [\mathbf{Z}^* \mathbf{c}_x](\eta)$. Thus we can simplify to:

$$p(x) \cdot \|f_x - \mathbf{Z}^* \mathbf{c}_x\|_q^2 \leq \tau_{s,p}(x) \cdot \int_{\eta \in \mathbb{R}} \int_{x \in \mathbb{R}} |f_\eta(x) - \mathbf{C}_s \mathbf{z}_\eta(x)|^2 p(x) q(\eta) dx d\eta$$

$$= \tau_{s,p}(x) \cdot \int_{x \in \mathbb{R}} \|f_x - \mathbf{Z}^* \mathbf{c}_x\|_q^2 p(x) dx$$

$$= \tau_{s,p}(x) \cdot \operatorname{tr}(\mathcal{K}_{p,q} - \mathbf{C}_s \mathbf{Z} \mathbf{Z}^* \mathbf{C}_s^*)$$

$$\leq \tau_{s,p}(x) \cdot 4\lambda s_{p,q,\lambda},$$

where the last two bounds follow from Theorem 31. □

□

## C.3 Active regression bounds

We conclude by combining the leverage score sampling result of Theorem 28, and Claim 29 with the kernel operator leverage score upper bound of Theorem 30 to solve Problem 6 with sample complexity depending polynomially on the statistical dimension $s_{p,q,\lambda}$.

**Corollary 33** (Active Function Fitting – Gaussian or Exponential Density)**.** *Consider the active regression set up of Problem 6. Let $p$ be the Gaussian density $p(x) = \frac{1}{\sigma \sqrt{2\pi}} e^{-x^2/(2\sigma^2)}$.*

*For any frequency density $q$ and $0 < \lambda < \|\mathcal{K}_{p,q}\|_{\mathrm{op}}$, let $s_{p,q,\lambda}$ be the $\lambda$-statistical dimension of $\mathcal{K}_{p,q}$. Let $s = \lceil 36 s_{p,q,\lambda} \rceil + 1$ and let $\bar{\tau}_{s,p}(x)$ be the leverage score bound of Theorem 1. Let $m = c \cdot s_{p,q,\lambda}^{5/2} \cdot (\log s_{p,q,\lambda} + 1/\delta)$ for a sufficiently large constant $c$. Let $x_1, \ldots, x_m$ be time points sampled independently according to the density proportional to $\bar{\tau}_{s,p}(x)$ and let $\tilde{y}$ be computed from these points using kernel ridge regression according to the procedure of Theorem 28 and Claim 29.*

*Then with probability $\geq 1 - \delta$:*

$$\|y - \tilde{y}\|_p^2 + 6\lambda \|g\|_q^2 + 8\|n\|_p^2. \tag{30}$$

*An identical bound holds when $p$ is the Laplacian density $p(x) = \frac{1}{\sqrt{2}\sigma} e^{-|x|\sqrt{2}/\sigma}$, $\bar{\tau}_{s,p}(x)$ is the leverage score bound of Theorem 2, and $m = c \cdot s_{p,q,\lambda}^2 \cdot (\log s_{p,q,\lambda} + 1/\delta)$.*

**Universal Sampling.** We remark that the sampling distribution of Corollary 33 is *independent of the frequency density $q$*. That is, we can fit a wide range of Fourier constrained functions (bandlimited, multiband, Gaussian process with any underlying kernel, etc.) with a single *universal* sampling scheme. This is surprising and reflects the universality of Fourier sparse functions in approximating these classes of functions through the frequency subset selection result of Theorem 31.

**Achieving Optimal Sample Complexity.** The sample complexity bounds of Corollary 33 are polynomial in $s_{p,q,\lambda}$ rather than linear, as is essentially optimal. We note that a near linear bound could be obtained by simply applying a second sampling step to the final kernel ridge regression problem of Claim 29, using the ridge leverage scores of the finite kernel matrix $\mathbf{K}$ [Sar06, DMM06a]. This is analogous to the final finite-dimensional random projection employed in Section B.3. A full

proof requires an extension of Theorem 28, which applies to an approximate solution of the finite ridge regression problem. This extension was shown in [AKM$^+$19].

Alternatively, it may be possible to improve our bounds on the kernel operator leverage scores (Def. 7). In [AKM$^+$19] sample complexity $O(s_{p,q,\lambda} \log s_{p,q,\lambda})$ is shown when $p$ is the uniform density over an interval. This proof starts from a bound essentially equivalent to Theorem 30. It then tightens this bound via a shifting argument that bounds the kernel leverage scores of $x$ near the edge of the interval with the leverage scores of $x$ closer to the center. It is not immediately clear how to extend such an argument to the case when $p$ is the Gaussian or Laplace density, but we believe than doing so may be possible. In general, we conjecture that a simple closed form leverage score bound that achieves within a constant factor of the optimal sample complexity exists.

## D   Empirically Estimating the Leverage Scores

The main technical challenge of this paper is to prove rigorous upper bounds on the leverage scores of a function class $\mathcal{F}$, under a distribution $p$. To do so, it is useful to have a way of empirically estimating the *true* leverage function $\tau_{\mathcal{F},p}$. Such an estimate may not be accurate for all $x$, and it may not have a closed-form. However, a good enough estimate can serve as guidance in proving theoretically sound bounds.

For some function classes (e.g., low-degree polynomials) establishing an empirical estimate for $\tau_{\mathcal{F},p}(x)$ is straight-forward. The class of sparse Fourier functions, $\mathcal{T}_s$, studied in this paper presents a somewhat greater challenge, but we are able to obtain relatively good estimates, including those used to plot Figure 1. In this section we briefly discuss our approach, which might be useful for future work, for example on other distributions beyond Gaussian and Laplace. MATLAB code for reproducing Figure 1 can be found in `empirical_upper_bounds.m` of the supplemental.

The key observation is that the function class $\mathcal{T}_k$ is a union of linear subspaces, and for each subspace, it is possible to relatively easily approximate the true leverage scores. In particular, for any *fixed* choice of frequencies $\lambda_1, \ldots, \lambda_k \in \mathbb{R}$, consider the function class:

$$\mathcal{T}_{\lambda_1,\ldots,\lambda_k} = \left\{ f : f(x) = \sum_{j=1}^{k} a_j e^{i\lambda_j x}, a_j \in \mathbb{C} \right\}.$$

For any fixed set of frequencies, $\mathcal{T}_{\lambda_1,\ldots,\lambda_k}$ is a subset of $\mathcal{T}_k$ and

$$\mathcal{T}_k = \bigcup_{\lambda_1,\ldots,\lambda_k \in \mathbb{R}} \mathcal{T}_{\lambda_1,\ldots,\lambda_k}.$$

So, if we let $\tau_{\lambda_1,\ldots,\lambda_k,p}(x)$ denote the leverage score of $\mathcal{T}_{\lambda_1,\ldots,\lambda_k}$, then the leverage scores of $\mathcal{T}_k$ equal:

$$\tau_{k,p}(x) = \sup_{\lambda_1,\ldots,\lambda_k \in \mathbb{R}} \tau_{\lambda_1,\ldots,\lambda_k,p}(x). \tag{31}$$

This equation is useful because, for any fixed $\lambda_1, \ldots, \lambda_2$, the right hand side is actually relatively easy to approximate. In particular, any function $f$ in $\mathcal{T}_{\lambda_1,\ldots,\lambda_k}$ can be written as $\mathcal{A}\alpha$ where $\alpha \in \mathbb{C}^k$ and $\mathcal{A}$ is an infinite dimensional linear operator with $k$ columns, the $j^{\text{th}}$ being equal to $e^{i\lambda_j x}$. I.e., $\mathcal{T}_{\lambda_1,\ldots,\lambda_k}$ is a $k$ dimensional linear subspace. If we are estimating the leverage scores with respect to distribution $p$, let $\bar{\mathcal{A}}_p$ be the rescaled linear operator with $j^{\text{th}}$ column equal to $e^{i\lambda_j x}\sqrt{p}$. We have

$$\tau_{\lambda_1,\ldots,\lambda_k,p}(x) = \sup_{\alpha \in \mathbb{C}^k} \frac{|\bar{\mathcal{A}}_p \alpha(x)|^2}{\|\bar{\mathcal{A}}_p \alpha\|_2^2}. \tag{32}$$

It is well know that the optimal $\alpha$ for maximizing (32) can be obtain by setting $\alpha = (\bar{\mathcal{A}}_p^* \bar{\mathcal{A}}_p)^{-1} \bar{\mathcal{A}}_p(x)$ where $\bar{\mathcal{A}}_p^*$ is the adjoint operator of $\bar{\mathcal{A}}_p$ [AKM$^+$17, Bac17, AKM$^+$19]. This leads to a leverage score of $\tau_{\lambda_1,\ldots,\lambda_k,p}(x) = \bar{\mathcal{A}}_p(x)^* (\bar{\mathcal{A}}_p^* \bar{\mathcal{A}}_p)^{-1} \bar{\mathcal{A}}_p(x)$, where $\bar{\mathcal{A}}_p(x)^*$ is the conjugate transpose of the $k$ length vector $\bar{\mathcal{A}}_p(x)$. While these expression involves infinite dimensional operators indexed by values in $\mathbb{R}$, they can be very well approximated for any $x$ discretizing $\bar{\mathcal{A}}_p$ to a finite number of rows. Specifically, $\bar{\mathcal{A}}_p$ is replaced with a matrix $\bar{A}_p$ with rows indexed $t \in \{-R, -R+\Delta, -R+$

$2\Delta,\ldots,R-\Delta,R\}$, each equal to $\left[e^{i\lambda_1 t}\sqrt{p(t)/\Delta} \quad \ldots, \quad e^{i\lambda_k t}\sqrt{p(t)/\Delta}\right]$ and we can approximate $\alpha \approx (\bar{A}_p^*\bar{A}_p)^{-1}\bar{A}_p(x)$ for any given $x$. The leverage score is approximated as $\tau_{\lambda_1,\ldots,\lambda_k,p}(x) \approx \bar{A}_p(x)^*(\bar{A}_p^*\bar{A}_p)^{-1}\bar{A}_p(x)$

With these equations in hand, our full approach for estimating $\tau_{k,p}(x)$ for a given $x$ is:

- Set $\tau_{k,p}(x) = 0$.
- For $iter = 1,\ldots,N$
    - Randomly select $k$ frequencies $\lambda_1,\ldots,\lambda_k \in \mathbb{R}$.
    - Approximately compute $\tau_{\lambda_1,\ldots,\lambda_k,p}(x)$ via discretization.
    - Set $\tau_{k,p}(x) = \max(\tau_{k,p}(x), \tau_{\lambda_1,\ldots,\lambda_k,p}(x))$.

To ensure this approach obtains a good approximation, it is important that the method for randomly selecting subsets of $k$ frequencies provides good "coverage", as different frequency subsets can lead to very different values of $\tau_{\lambda_1,\ldots,\lambda_k,p}(x)$. One point to note is that, as frequencies become far apart, the columns of $\bar{\mathcal{A}}_p$ become close to mutual orthogonal, and the leverage scores converge to the squared $\ell_2$ norms of the rows of $\bar{\mathcal{A}}_p$, which equal $k \cdot p(x)$ for any $x$. This means that the benefit of considering subsets involving distance frequencies is marginal, as such subsets always lead to approximately the same scores. So, we can focus on sampling values of $\lambda_1,\ldots,\lambda_k$ that are relatively close together.

To generate the plots of Figure 1, we do so via independent sampling. At each iteration, a random order of magnitude $h$ was chosen on a geometric grid between .01 and 10 and $\lambda_1,\ldots,\lambda_k$ where chosen as random Gaussians with variance $h$. A large number of iterations (10 million) was run, and the range of $h$ was increased until doing so had no noticeable effect on the estimate for $\tau_{k,p}(x)$. This leaves us reasonable confident that the curves of Figure 1 accurately reflect the true leverage scores, although we of course can not be sure, as the method is only heuristic.

## E   Details for Experiments

**Details of Sampling.** MATLAB code for our modified random Fourier features method is included in `gaussianKernelMRFF.m` and `cauchyKernelMRFF.m`. It uses simplified versions of the leverage score upper bounds from Theorems 1 and 2. In particular, in both of these theorems, the leverage score upper bound distributions are piecewise, following a different functions for frequencies above and below a certain cutoff $F$. $F$ equals $6\sqrt{2}\sigma \cdot \sqrt{s}$ and $9\sqrt{2}\sigma \cdot s$ in Theorems 1 and 2, respectively. The bulk of each distribution is on values of $|\eta| \leq F$, so we ignore the "tail" part of each distribution when sampling. This does not seem to significantly effect the experimental results. More over, using the empirical leverage score score distributions from Figure 1 as guidance, we used tighter values for $F$ than we were able to prove theoretically. For example, setting $F = 4\sigma$ seems sufficient to capture the bulk of the Fourier sparse leverage score distribution for the Gaussian measure, so this is the value we used in our experiments. I.e. samples of $\eta$ were drawn uniformly from the ball $\{\eta : \|\eta\|_2 < 4\sigma\}$.

When sampling we also use the same trick from [RR07] to achieve a real valued embedding, which makes it easier to work with the embedding downstream (e.g., when implementing the preconditioned solver). In particular, instead of including $C \cdot e^{-2\pi i\eta^T x}$ in the embedding, where $C$ is the appropriate constant as in Definition 3, we can include an entry equal of $C \cdot \cos(2\pi\eta^T x + \beta)$ where $\beta$ is a uniform random variable from $[0, 2\pi]$. It's not hard to check that the corresponding real valued embedding will still satisfy $\mathbb{E}[\mathbf{G}^*\mathbf{G}] = \mathbf{K}$, and experimentally, approximation quality does not appear to suffer.

**Details of Preconditioning.** When solving $(\mathbf{K} + \lambda\mathbf{I})^{-1}\mathbf{z}$ with a preconditioner, each iteration of the preconditioned solver requires 1) computing $(\tilde{\mathbf{K}} + \lambda\mathbf{I})^{-1}\mathbf{z}$ for some vector $\mathbf{z}$ and 2) multiplying $\mathbf{K} + \lambda\mathbf{I}$ by a vector $\mathbf{w}$. The first step can be done efficiently whenever $\tilde{\mathbf{K}} = \mathbf{G}^*\mathbf{G}$ where $\mathbf{G} \in \mathbb{C}^{n\times m}$, which is the type of approximation we get from a random Fourier features method. In particular, let $\mathbf{G} = \mathbf{U}\boldsymbol{\Sigma}\mathbf{V}^T$ be $\mathbf{G}$'s singular value decomposition. Due to the simplification discussed above, $\mathbf{G}$ is always real-valued in our setting, and so is its SVD. We have $\mathbf{U} \in \mathbb{R}^{m\times m}$, $\boldsymbol{\Sigma} \in \mathbb{R}^{m\times m}$, and $\mathbf{V} \in \mathbb{R}^{n\times m}$. The SVD can be computed in $O(m^2 n)$ time and more importantly, this operation is very fast when $\mathbf{G}$ fits in memory, which is often possible even when $\mathbf{K} \in \mathbb{R}^{n\times n}$ does not. So, for both classical RFF preconditioning and modified RFF preconditioning, we choose values for $m$ that allow for fast computation of the SVD, and compute the decomposition as a preprocessing step.

Then, it's not hard to check that $(\tilde{\mathbf{K}} + \lambda\mathbf{I})^{-1}\mathbf{z} = \mathbf{V}(\boldsymbol{\Sigma} + \lambda\mathbf{I}_{m\times m})^{-1}\mathbf{V}^T\mathbf{z} + \frac{1}{\lambda}(\mathbf{z} - \mathbf{V}^T\mathbf{z})$, which can be computed in $O(mn)$ time. This is much faster than the cost of multiplying a vector by $\mathbf{K} + \lambda\mathbf{I}$, so the cost of preconditioning ends up being a lower order term in the solver complexity: it increases the cost of each iteration by just a small factor.