[Reviews · NeurIPS 2020]

Review 1

Summary and Contributions: This submission concerns kernel-based learning methods, which have been well studied in machine learning. Methods such as Random Fourier Features and the Nystrom method have been used to scale up kernel methods by computing kernel approximations. Such methods generally fall into two categories, namely, oblivious methods (e.g., RFF, Tensor Sketch variant) and data-adaptive methods (Nystrom method). While the former have the advantage of parallelizability (in settings where data is distributed across different machines), the latter have generally given better kernel approximations until now. This work closes this gap between the two classes of methods by advancing an oblivious sketching method with better approimation, based on statistical leverage scores. The authors also show how to use the method in an active learning setting. Statistical leverage scores have gained increasing importance in literature on kernel methods. For instance, Avron et al. (ICML'17) were able to upper bound leverage scores in the case of the RBF Gaussian kernel on bounded data sets in low dimension; they showed that this is enough to obtain better kernel approximations for problems such as ridge regression with sample dependence polynomial in the statistical dimension. This work roughly follows this line of thought, except that tighter upper bounds on the leverage score are produced without the assumption of bounded data sets. Furthermore, experiments are provided on a GIS dataset, for which a comparison of the modified random Fourier features method (via statistical leverage score sampling according to the theory in this paper) to the classical RFF method is provided. While classical RFF has better error, the new RFF is shown to approximate small eigenvalues of the kernel matrix better.

Strengths: The problem is well motivated--finding optimal oblivious sketches has appeared as an important open question in works such as Avron et al. (ICML'17) and Ahle et al. (SODA'20). The authors seemingly make progress toward this important problem for Gaussian and Laplace kernels. In the case of Gaussian kernels, this work has the added advantage over the aforementioned works of not requiring an assumption on boundedness of the data. The work has solid theoretical grounding (though there are some questions I have, see below section), and the theory is supplemented by reasonable experiments on a real-world GIS dataset.

Weaknesses: Unlike the work of Ahle et al. (which gives polynomial dependences for high-dim poly and Gaussian kernels), the work only seems to apply to the case of data with low ambient dimension. It would be good to give some more intuition as to why this is the case. Moreover there are some questions about the theory that are unclear (see questions below). The experiments display a tradeoff between error and approximation of small eigenvalues. The normal RFF method does better in the former. It's not clear to me why better approximation of small eigenvalues at the expense of overall error is a suitable tradoeff. The authors mention preconditioning/spectral approximation as a motivation, but it is not clear to me how much this matters in practice. Some experiments that perform linear algebra based on preconditioning methods would offer more convincing evidence of the superiority of the authors' approach.

Correctness: There are a few claims and applications of theorems that I could not follow, but it is possible I am missing some insight here (see below questions).

Clarity: Yes, the paper is written in a clear, easy-to-read fashion.

Relation to Prior Work: Yes, the work does a good job of motivating the work in the context of previous literature.

Reproducibility: Yes

Additional Feedback: I have some questions regarding some of the proofs, which impact correctness of the paper's main theoretical contributions. I am willing to upgrade my overall score for the paper subject to satisfactory responses to these questions. (1) How does the equation after line 518 follow? Lemma 7 bounds the L2 norm on a smaller interval, whereas the equation is trying to upper bound the L-infinity norm. In general, it is not the case that the L-infinity norm of f is upper bounded by the L2 norm on the described interval (i.e., |f(x)| <= ||f||_[a-delta, b+delta]), so the logic seems incorrect. (The same appears to be true for the logic in the equation following line 519, which is symmetric.) (2) I could not find Lemma 7 in [BE06]. Again, there appears to be a mismatch between L2 and L-inf norms. Do the authors have a specific citation? [UPDATE: POST-REBUTTAL] Thanks for the detailed responses to my questions. I am convinced of the correctness now and am willing to upgrade the review to a 7. This is a good submission.


Review 2

Summary and Contributions: This paper provides an improved theoretical bound for upper bounding leverage scores conditioned on representing s-sparse Fourier functions. This is particular relevant for various kernel learning applications. The paper describes how these result in improved results for kernel learning applications (e.g., kernel regression) when the bound depends on the statistical dimension, and the sketches are oblivious. In this case, the dimension of an approximate lifting map (e.g., like Rahimi-Recht Random Fourier Features) can achieve *relative* error in the kernel-ridge problem, using dimensions only depending on the relative error parameter and the statistical dimension. This is for instance what is required for approximate versions of kernel ridge regression.

Strengths: These are important theoretical results, for problems quite central to NeurIPS, and thus should be accepted. The paper also provides a nice array of empirical uses cases. These basically are the same as recent non-oblivious ways to featurize the data (beyond Rahimi-Recht RFFs), and they show clear improvement. This paper shows these results are oblivious to the data (like RR RFFs), which is important for big data settings.

Weaknesses: The only complaint is there is still a sqrt{s} gap in the leverage score bound for Gaussian densities, that seems significant (upper bound is Omega(s^{3/2}), lower bound is s). The experiments mentions that in practice the algorithm (from how it is implemented is not really new), but that is fine, as the main importance of this work is showing theoretical result and obliviousness.

Correctness: It all seems correct. Proofs are all sketched in main 8 pages.

Clarity: Yes!

Relation to Prior Work: Note: The formulation of leverage scores (in equation (1)) is indeed a nice general view. In fact, it basically appears like the notion of total sensitivity (https://doi.org/10.1137/1.9781611973075.50); these concepts are known to be the same in many cases, so it may be useful to make this connection explicit.

Reproducibility: Yes

Additional Feedback:


Review 3

Summary and Contributions: The authors derive a computationally tractable upper-bound on the leverage score function when the function class is that of Fourier sparse functions with s spectrums and the probability density is that of Gaussian or Cauchy, in the univariate setting. They use the derived bound in two applications: random features for kernel approximation and active sampling.

Strengths: - The paper contains a lot of interesting theoretical results. I found in particular Theorem 5 is interesting, as it connects the \lambda-ridge leverage score in Def 4 to the leverage score function in Eq.1 defined with the Fourier-sparse function class and the spectral density of the kernel. This is an interesting result in its own right. Comined with the computationally tractable upper bound in Theorem 1 or Theorem 2, this leads to a tractable approximation to the \lambda-ridge score for sampling random features. - The presentation is good, as there is a coherent story. I enjoyed reading it. - The theoretical results would be interesting to kernel folks.

Weaknesses: - The theoretical results only hold for the univariate case with Gaussian or Cauchy densities. - The proposed approach to sampling random features implicitly assumes that the \lambda-statistical dimension in Def 2 is known. This can be seen from Theorem 5, whose upper-bound includes the statistical dimension. Since the statistical dimension requires the computation of the eigenvalues of the kernel matrix, this requires O(n^3) computational cost, if exactly computed. Since this cost is what we want to avoid (that's why we use random features), this makes the proposed approach practically useless. Therefore the authors should discuss this point and explain if there is any computationally efficient way of approximating the statistical dimension. - The experimental setting is not explained in detail, and the experiments may not be perfomed systematically. Regarding the former, the authors do not explain how the number of spectrum s (or the corresponding statistical dimension s_\lambda) is specified and computed, for instance. Regarding the latter, the results do not have error bars, i.e., standard deviations obtained from independent experiments.

Correctness: I have not checked the proofs in the supplementary. Some important details are missing in the experimental results (see Weaknesses above).

Clarity: Yes, the paper is clearly written in general. However, there are some places where the explanations are not so clear. - Theorem 3: what is the form of the embedding? The theorem only claims its existence. This is important as this embedding is used as a proposed method (lines 251-252) - Line 197: what is q? - Lines 251-252: "To achieve the linear dependence on s_\lambda in Theorem 3 we argue that it suffices to post-process the modified RFF embedding g with a standard random projection method". It is not very clear from the text what this means. Do you have any formal statement of this and a formal theoretical justification for it?

Relation to Prior Work: Yes One paper to be discussed: the following paper seems closely related to this paper. Please discuss how your work is related to their analysis. Z. Li, J.-F. Ton, D. Oglic, and D. Sejdinovic, Towards A Unified Analysis of Random Fourier Features, in International Conference on Machine Learning (ICML), 2019, PMLR 97:3905–3914

Reproducibility: No

Additional Feedback: *** Post rebuttal *** Thanks for your feedback!


Review 4

Summary and Contributions: This paper first provides two upper bounds on the leverage scores of Fourier sparse functions under Gaussian and Laplace measures. The authors motivate computing these upper bounds by the fact that for these functions, it is difficult to compute the leverage score over all x in the domain. To demonstrate usefulness of the obtained bounds, the authors talk about two ML applications. First, they show that the bounds yield a modified random Fourier features algorithm that can approximate Gaussian and Cauchy kernel matrices. Specifically, they show that for low dimensional data, a data oblivious kernel approximation method can be used that nearly matches best adaptive methods in speed and approximation quality. Second, they show that the bounds can be used as non-uniform sampling distributions for active learning. Specifically, they obtain active sampling results for regression problems involving s arbitrary complex exponentials when the data follows a Gaussian or Laplacian distribution. Finally, to show the potential of Fourier sparse leverage score bounds, the proposed modified RFF method is empirically evaluated and compared with classical RFF on a kernel ridge regression problem. They show that their method leads to faster convergence and better prediction error.

Strengths: The authors highlight the importance of their work, their contributions, and relation to prior work well. Their numerical section is also detailed.

Weaknesses: In general, while the authors make interesting claims, there are some issues with the work: Section 2: After both Theorem stating the leverage scores upper bounds, the authors mention that there may be a gap and elude to the fact that a tighter upper bound may be obtained by strengthening Theorem 1 and 2. Do the authors have any ideas of how their Theorems can be strengthened? O(\sqrt(s)) does seem to be a large gap in Theorem 1. I saw in the Appendix that the authors do give a tight bound for a special case. This should be called out in the main text and more explanations have to be given there. I found the fact that illustrated plots related to these upper bounds were also “tighter” than what authors can prove a bit misleading. I think it’s better to plot the actual obtained upper bound. Page 3 line 122 an “=” is missing? Section 3: The authors mention “low dimensional data” here and in abstract. I think it’s better if that is quantified orderwise. Also the naming of subsections 3.1 and 3.2 is misleading. Theorem 3 is the authors contribution in subsection 3.1 (Formal results). The next subsection is named “Our approach” which usually means that the previous subsection was prior work. Section 4: The authors mention that they use kernel approximation as a preconditioner to accelerate the iterative solution of the original problem. They say that they get similar empirical results as [AKM+17] did by using approximation in place of K. Is this similarity in terms of accuracy or speed? If both, it’s not clear what makes this algorithm interesting for this use-case. Update: The authors did address this point in rebuttal.

Correctness: I did not check the proofs of Theorems. Empirical methodology looks correct.

Clarity: Overall the paper is well written. The flow can be improved. I found myself reading parts more than once and going back and forth between Theorems.

Relation to Prior Work: The authors do make distinctions between their contributions and prior work throughout the paper.

Reproducibility: Yes

Additional Feedback:

[Author Response · NeurIPS 2020]

We thank the reviewers for their helpful suggestions and clarifying questions. In summary, our work makes theoretical
progress on the challenging and practically important question of oblivious kernel embeddings. It leaves open some
interesting questions which we would be excited to work on or see solved by other researchers. As the reviewers note,
the extra $\sqrt{s}$ in our Gaussian kernel upper bound and extensions to higher dimensions are particularly enticing.

Responses to specific points are below.

**Extension to higher dimensions:** Our results can be directly extended to higher dimensions, but with an exponential
in dimension cost. We believe this is unnecessary, and proving such a result is a major open question. [AKK+20b]
gives a dimension independent result, but with an additional dependence on the data set radius. The goal would be to
remove the radius dependence and only depend on the statistical dimension $s_\lambda$.

**Reviewer 1:**

*The authors mention preconditioning/spectral approximation as a motivation...some experiments that perform linear*
*algebra based on preconditioning methods would offer more convincing evidence...*

See Figure 4a. We show faster convergence in solving a standard kernel ridge regression problem with preconditioned
CG based on our modified RFF method vs. the traditional RFF method.

*How does the equation after line 518 follow? Lemma 7 bounds...*

Thanks very much for pointing out this confusion. We will clarify the discussion and some typos in the paper.

First, note that Lemma 6 is given as Theorem 7.1 in [Erd17] with an additional factor 2 in it. This factor does not
significantly affect any results, and thus we could just use this theorem directly. We also seemed to have dropped a
factor of two from [BE06] in Lemma 7 – thus, in retrospect we should have just cited [Erd17] in the first place.

To clarify our own proof: as pointed out by the reviewer, Lemma 7 should be stated with $\|f\|_{L_\infty[a+\delta,b-\delta]}$ in place of
$\|f\|_{[a+\delta,b-\delta]}$ as in [BE06]. Then, to prove Lemma 6 (focusing on the first bound since the second is symmetric), we set
$\delta = b - x$ and thus have:

$$\frac{|f(x)|^2}{\|f\|_{[a,b]}^2} \leq \frac{\|f\|_{L_\infty[a+b-x,x]}^2}{\|f\|_{[a,b]}^2} = \frac{\|f\|_{L_\infty[a+\delta,b-\delta]}^2}{\|f\|_{[a,b]}^2} \leq \frac{s}{\delta} = \frac{s}{b-x}.$$

Note that in the first inequality, we use $a+b-x \leq x$ which follows from the assumption in this case that $x \geq (a+b)/2$.

**Reviewer 2:** We hope that the $\sqrt{s}$ gap in the Gaussian density leverage score bound can be closed. This is an exciting
open question. Note that prior to our work, no polynomial in $s$ bound was known.

**Reviewer 3:**

*The proposed approach to sampling random features implicitly assumes that the $\lambda$-statistical dimension is known...*

This is a good point – in theory $s_\lambda$ needs to be known. In practice simple heuristics seem to suffice – see Appendix E of
the supplement for more details on our implementation.

*Theorem 3: what is the form of the embedding? The theorem only claims its existence...*

See Corollary 27 in the appendix for a full statement of Theorem 3, which gives the explicit random features construction.
This also explicitly discusses and formally proves the post-processing via random projection step, which the reviewer
asks about. We will try to expand discussion in the main body, subject to space constraints.

In regards to the connection with Towards A Unified Analysis of Random Fourier Features: our results can be directly
plugged into that work, which nicely applies to any method that samples random features with an upper bound on the
ridge leverage scores. That work gives a leverage score approximation method but 1) it is not oblivious and 2) it has a
$1/\lambda$ dependence, which for small $\lambda$ can be worse than our $s_\lambda$ dependence.

**Reviewer 4:** In regards to obtaining tighter upper bounds, see our response to Reviewer 2. We do not yet have an
approach to closing the $O(\sqrt{s})$ gap but are working on it.

*The authors mention that they use kernel approximation as a preconditioner to accelerate the iterative solution of the*
*original problem. They say they get similar empirical results as [AKM+17] did by using approximation in place of K...*

To clarify, our embedding method (not results) are very similar to that of [AKM+17]. Due to this similarity, we test a
different method to solve linear systems, using a preconditioner rather than a direct approximation (as AKM+17 already
tests the direct approximation approach). These methods are somewhat incomparable. Preconditioning leads to much
higher accuracy at the expense of possibly slower runtimes when $n$ is large. Either embedding method can be used with
either system solving approach.

[Meta-Review · NeurIPS 2020]

Four knowledgeable reviewers recommend to accept the paper based on the progress it has made toward understanding and improving the accuracy of kernel approximations made using leverage scores. On this basis the paper is accepted. We ask that the authors implement the edits proposed in their rebuttal: to clarify the justification for the equation after line 518, and to expand on the discussion of Theorem 3.